# Cloud top pressure retrieval with DSCOVR-EPIC oxygen A and B bands observation

Bangsheng Yin[1], Qilong Min[1, *], Emily Morgan[1], Yuekui Yang[2], Alexander Marshak[2], and Anthony B. Davis[3]

[1]Atmospheric Sciences Research Center, University at Albany, Albany, NY, USA

[2]NASA Goddard Space Flight Center, Climate and Radiation Laboratory, Greenbelt, MD, USA

[3]Jet Propulsion Laboratory, California Institute of Technology, Pasadena, CA, USA

[*]Corresponding author, qmin@albany.edu

**Abstract**

An analytic transfer inverse model for Earth Polychromatic Imaging Camera (EPIC) observation is proposed to retrieve the cloud top pressure (CTP) with considering in-cloud photon penetration. In this model, an analytic equation was developed to represent the reflection at top of atmosphere from above cloud, in-cloud, and below-cloud. The coefficients of this analytic equation can be derived from a series of EPIC simulations under different atmospheric conditions using a non-linear regression algorithm. With estimated cloud pressure thickness, the CTP can be retrieved from EPIC observation data by solving the analytic equation. To simulate the EPIC measurements, a program package using the double-$k$ approach was developed. Compared to line-by-line calculation, this approach can calculate high-accuracy results with a one-hundred-fold computation time reduction. During the retrieval processes, two kinds of retrieval results, i.e., baseline CTP and retrieved CTP, are provided. The baseline CTP is derived without considering in-cloud photon penetration, and the retrieved CTP is derived by solving the analytic equation, taking into consideration the in-cloud and below-cloud interactions. The retrieved CTP for the oxygen A and B bands are smaller than their related baseline CTP. At the same time, both baseline CTP and retrieved CTP at the oxygen B-band are larger than those at the oxygen A-band. Compared to the difference of baseline CTP between the B-band and A-band, the difference of retrieved CTP between these two bands is generally reduced. Out of around 10000 cases, in retrieved CTP between A- and B-bands we found an average bias of 93 mb with standard deviation of 81 mb. The cloud layer top pressure from Cloud-Aerosol Lidar and Infrared Pathfinder Satellite Observations (CALIPSO) measurements is used to do validation. Under single-layer cloud situations, the retrieved CTPs for the oxygen A-band agree well with the CTPs from CALIPSO, which mean difference is within 5 mb in the case study. Under multiple-layer cloud situations, the CTPs derived from EPIC measurements may be larger than the CTPs of high level thin-clouds due to the effect of photon penetration.

## 1.  Introduction

The Deep-Space Climate Observatory (DSCOVR) satellite is an observation platform orbiting within the first Sun-Earth Lagrange point (L1), 1.5 million km from the Earth, carrying a suite of instruments oriented both Earthward and sunward. One of the Earthward instruments is the Earth Polychromatic Imaging Camera (EPIC) sensor, which can take images of the Earth with spatial resolution of 10 km at nadir. The EPIC continuously monitors the entire sunlit Earth for backscatter, with a nearly constant scattering angle between 168.5º and 175.5º, from sunrise to sunset with 10 narrowband filters: 317, 325, 340, 388, 443, 552, 680, 688, 764 and 779 nm (Marshak et al., 2018). Of the 10 narrow-band channels, there are two oxygen absorption and reference pairs, 764nm versus 779.5nm and 680nm versus 687.75nm, for oxygen A and B bands. The cloud top pressure (CTP) or cloud top height (CTH) is an important cloud property for climate and weather studies. Based on differential oxygen absorption, both EPIC oxygen A-band and B-band pairs can be used to retrieve CTP. It is worth noting that although CTP and CTH reference the same characteristic of clouds, the conversion between the two depends on the related atmospheric profile.

Although the theory of using oxygen absorption bands to retrieve CTP was proposed decades ago (Yamamoto and Wark, 1961), it is still very challenging to do the retrieval accurately due to the complicated in-cloud penetration effect (Yang et al., 2019, 2013; Davis et al., 2018a, 2018b; Richardson and Stephens, 2018; Loyola et al., 2018; Lelli et al., 2014, 2012; Schuessler et al., 2013; Rozanov and Kokhanovsky, 2004; Kokhanovsky and Rozanov, 2004; Kuze and Chance, 1994; O'brien and Mitchell, 1992; Fischer and Grassl, 1991; and etc.). To estimate the CTP from satellite measurements, many approaches have been designed to retrieve clouds' effective top pressures without considering in-cloud photon penetration. These approaches did not consider light penetrating cloud, therefore the derived CTH is lower than the cloud top,  and the effective top pressures is higher than CTP. In the meantime, to improve the retrieval accuracy of CTP, various techniques have been applied to the retrieval methods with in-cloud photon penetration. For example, Kokhanovsky and Rozanov (2004) proposed a simple semi-analytical model for calculation of the top-of-atmosphere (TOA) reflectance of an underlying surface-atmosphere system, accounting both for aerosol and cloud scattering. Based on the work of Kokhanovsky and Rozanov (2004), Rozanov and Kokhanovsky (2004) developed an asymptotic algorithm for the CTH and the geometrical thickness determination using measurements of the cloud reflection function. This retrieval method was applied by Lelli et al. (2012, 2014) to derive CTH using measurements from GOME instrument on board the ESA ERS-2 space platform.

Currently, based on the measurements of DSCOVR EPIC sensor, the Atmospheric Science Data Center (ASDC) at National Aeronautics and Space Administration (NASA) Langley Research Center archives both calibrated EPIC reflectance ratio data and processed Level 2 cloud retrieval products, including cloud cover, cloud optical depth (COD), cloud effective top pressure at oxygen A and B bands (Yang et al., 2019). By using EPIC reflectance ratio data at oxygen A-band and B-band absorption to reference channels, Yang et al (2013) developed a method to retrieve CTH and cloud geometrical thickness simultaneously for fully cloudy scene over ocean surface. First their method calculates cloud centroid heights for both A- and B-band channels using the ratios between the reflectance of the absorption and reference channels, then derives the CTH and the cloud geometrical thickness from the two dimensional look up tables that relate the sum and the difference between the retrieved centroid heights for A- and B-bands

to the CTH and the cloud geometrical thickness. The difference in the $O_2$ A- and B-band cloud
centroid heights is resulted from the different penetration depths of the two bands. Compared to
the cloud height variability, the penetration depth differences are much smaller and the retrieval
accuracy from this method can be affected by the instrument noise (Davis et al. 2018a, b).
In this paper, to address the issue of in-cloud penetration, we proposed an analytic method
to retrieve the CTP by using DSCOVR EPIC oxygen A- and B-band observation. This analytical
method adopted ideas of the semi-analytical model (Kokhanovsky and Rozanov, 2004; Rozanov
and Kokhanovsky, 2004), and developed a quadratic EPIC analytic radiative transfer equation to
analyze the radiative transfer in oxygen A- and B-band channels. The structure of this paper is as
follows: section 2 describes the theory and methods, which includes several subsections, i.e., the
introduction of DSCOVR EPIC oxygen A and B bands filters, the theory of CTP retrieval based
on EPIC oxygen A- and B- band observation, and the detailed retrieval algorithm; section 3
describes the application and validation of the CTP retrieval method, which also includes several
subsections, i.e., case studies of CTP retrieval, validation of the retrieval method, and retrieval of
global observation; and section 4 states the conclusions of this study.

## 2.   Theory and methods

### 2.1 DSCOVR EPIC oxygen A and B bands filters

EPIC filters at 764 nm and 779 nm cover the oxygen A-band absorption and reference
bands, respectively (Fig. 1a). The high resolution absorption optical depth spectrum at oxygen A-
band and B-band is calculated by Line-By-Line Radiative Transfer Model (LBLRTM, Clough et
al., 2005) with HITRAN 2016 database (Gordon et al., 2017) for the U.S. standard atmosphere.
In this wavelength range, the O3 absorption is very weak (O3 optical depth < 0.003) and there
are no other gas absorptions. The background aerosol and Rayleigh scattering optical depth vary
smoothly within the A-band range; the differences between in-band and reference band are
negligible at nominal EPIC response functions. EPIC filters at 688 nm and 680 nm cover the
oxygen B-band absorption and reference band, respectively (Fig. 1b). Compared to the oxygen
A-band, O3 absorption is slightly stronger in the oxygen B-band range, with an O3 optical depth
around 0.01. Any water vapor absorption in the B-band range is negligible. In the standard
atmospheric model, from the oxygen B-band reference band to the absorption band, the O3
absorption and Rayleigh scattering optical depth decreased by approximately 0.002 and 0.002,
respectively. This may have some impacts on the CTP retrieval from the oxygen B-band (more
discussion in the later sections). It is worth noting that for EPIC measurements at both oxygen A-
and B-bands, the surface influence cannot be ignored. For examples, in the snow or ice covered
area the surface albedo is high; in the plants covered area, the surface albedo changes
substantially between oxygen A-band and B-band due to the impact of spectral red-edge (Seager
et al., 2005).

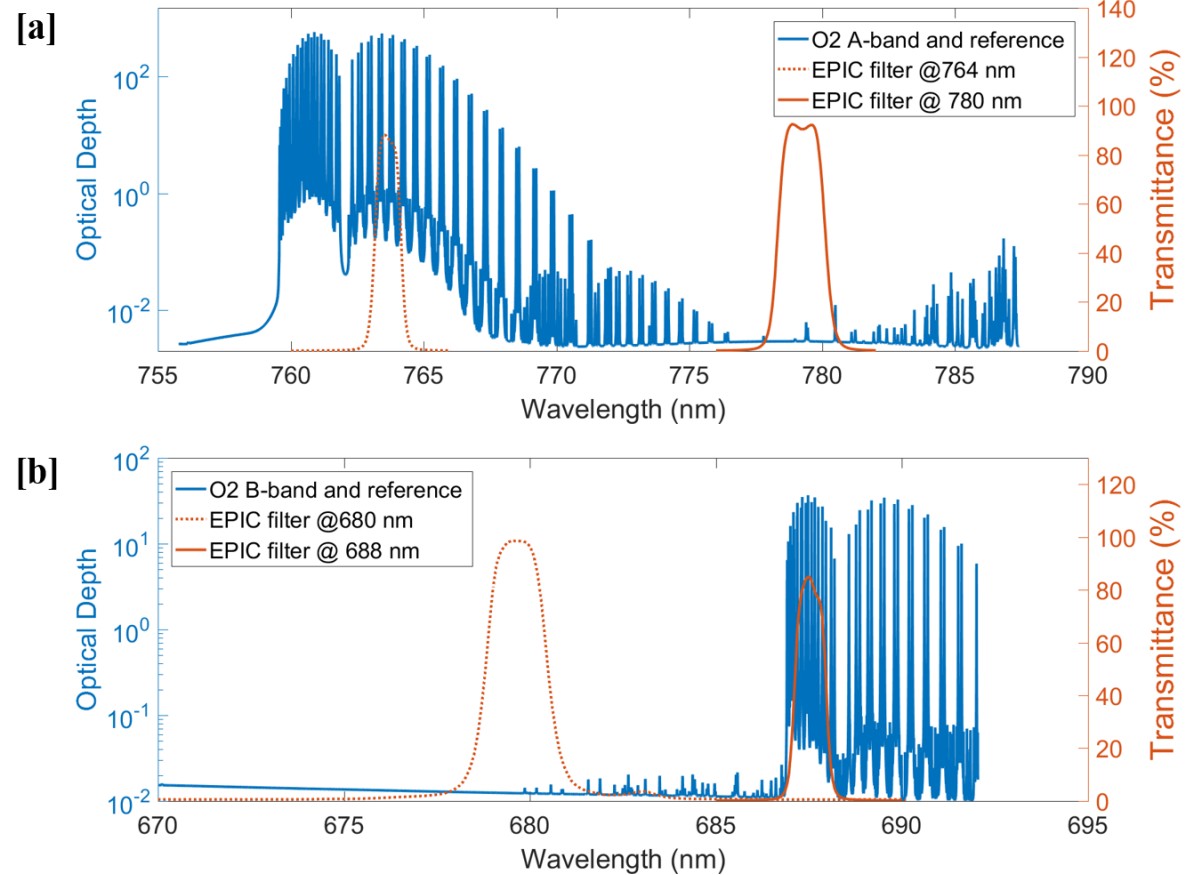

**Figure 1:** High resolution calculated absorption optical depth spectrum at oxygen A-band (a) and B-band (b) with DSCOVR EPIC oxygen A and B bands in-band and reference filters. Here the absorption optical depth spectrum is calculated by LBLRTM model with HITRAN 2016 database for the U.S. standard atmosphere.

In general, if we use the pair of oxygen A and B absorption and reference bands together, the impact of other absorption lines, background Rayleigh scattering, and aerosol optical depth are very limited. At the same time, as a well-mixed major atmospheric component, the vertical distribution of oxygen in the atmosphere is very stable under varying atmospheric conditions. Thus, we can use the ratio of reflected radiance (or reflectance) at the TOA of oxygen absorption and reference bands (i.e., $R_{764}$ and $R_{779}$, $R_{688}$ and $R_{680}$) to study the photon path length distribution and derive the cloud information. Also, compared to any specific EPIC oxygen absorption bands (i.e., $R_{764}$ and $R_{688}$), the ratios of absorption to reference channels (i.e., $R_{764}/R_{779}$ and $R_{688}/R_{680}$) are less impacted by the instrument calibration and other measurement error. This can be explained by the following reasons: First, the EPIC measurements at oxygen A and B absorption and reference bands share same sensor and optical system, when calculating the ratios of them, some preprocessing calibration errors can be reduced. Second, to calculate $R_{764}$ and $R_{688}$, the ratio of lunar reflectance at neighboring channels (i.e., $F(764,779)$ and $F(688,680)$) and the calibration factors of oxygen A and B reference bands (i.e., $K_{779}$ and $K_{680}$) are used (Geogdzhayev and Marshak, 2018; Marshak et al., 2018). Therefore, the accuracy of $R_{764}$ and $R_{688}$ is determined by the stability of $F(764,779)$

and $F(688,680)$ and the accuracy of $K_{779}$ and $K_{680}$ together. But the accuracy of absorption to
reference ratios is only determined by the stability of $F(764,779)$ and $F(688,680)$.

## 2.2 Theory of CTP retrieval based on EPIC oxygen A- and B- band observation

In our study, we tried two methods to retrieve the CTP based on EPIC oxygen A-band and
B-band measurements: (1) Build a lookup table (LUT) for various atmospheric conditions and do
the retrieval by searching the LUT; (2) Develop an analytic transfer inverse model for EPIC
observations and calculate the related coefficients based on a series of simulated values, then use
this analytic transfer inverse model to retrieve the CTP. In this paper, we mainly focus on the
second method.

### 2.2.1 Method 1: LUT based approach

One commonly used method of retrieval for satellite observation is through the building
and usage of LUTs (Loyola et al., 2018, Gastellu-Etchegorry and Esteve, 2003). LUT based
approach can be fast because the most computationally expensive part of the inversion procedure
is completed before the retrieval itself. For DSCOVR EPIC observations, we can build LUTs by
simulating EPIC measurements under various atmospheric conditions, such as different surface
albedo, solar zenith and viewing angles, COD, CTP, and cloud pressure thickness. Comparing
the related simulated reflectance at the oxygen absorption and reference bands, we can obtain
two LUTs for reflectance ratios of absorption/reference at EPIC oxygen A-band and B-band
respectively, which can be used for the CTP retrieval. The detailed information of simulated
reflectance ratio of absorption/reference is stated in Sect. 2.3.3.
During the retrieval process, the EPIC measurements (e.g., reflectance at oxygen A and B
bands) with related solar zenith and viewing angles can be obtained from the EPIC level 1B data;
COD information (retrieved from other EPIC channels) can be obtained from EPIC level 2 data.
At the same time, we can get surface albedo from Global Ozone Monitoring Experiment 2
(GOME-2) Surface Lambertian-equivalent reflectivity (LER) data (Tilstra et al., 2017). At this
point the CTP and cloud pressure thickness are the only unknown variables. The cloud pressure
thickness or the cloud vertical distribution has substantial impact on the accuracy of the CTP
retrievals (Carbajal Henken et al., 2015; Fischer and Grassl, 1991; Rozanov and Kokhanovsky,
2004; Preusker and Lindstrot, 2009). In this study, the cloud pressure thickness is used as an
input parameter to retrieve the CTP. However, no related accurate cloud pressure thickness is
provided by other satellite sensors now. To constrain the error from the estimation of cloud
pressure thickness, we related it to the cloud optical thickness. It is reasonable because clouds
with higher optical thickness normally have higher values of pressure thickness. To explore the
correlation between cloud pressure thickness and cloud optical thickness, we use the related
cloud data from Modern-Era Retrospective analysis for Research and Applications Version 2
(MERRA-2, Gelaro et al., 2017), which is a NASA atmospheric reanalysis for the satellite era
using the Goddard Earth Observing System Model Version 5 (GEOS-5) with Atmospheric Data
Assimilation System (ADAS). Based on statistical analysis of one year's single-layer liquid
water clouds over an oceanic region (23.20º S, 170.86º W, 2.11º S, 144.14º W) in 2017, we can
get an equation for cloud pressure thickness approximation, i.e., cloud pressure thickness (mb) =
2.5* COD + 23. The derived correlation coefficients are dependent on the case region and time
selections. Due to the complexity of cloud vertical distribution, whatever the accuracy of the
correlation coefficients is, the estimation will certainly bring in error.
With an estimated cloud pressure thickness, a multi-variable LUT searching method can
then be used to interpolate and obtain the CTP. It is worth noting that the reflectance ratio of
absorption/reference can be seen as a function of surface albedo, solar zenith and viewing angles,
COD, CTP, and cloud pressure thickness. Some atmospheric variables will have a non-linear
effect on the reflectance ratio. For example, the reflectance ratio is more sensitive to the variation
of COD when COD is small. Overall, the reflectance ratio varies monotonically and smoothly
with these variables (shown in Fig. 3). With a relatively high-resolution simulated table, we can
use a localized linear interpolation method to estimate the proper values. Multiple interpolations
are needed for this method to decrease the number of LUT dimensions, which will cost more
time than the analytic transfer inverse model method. The retrieval error of this method is
determined by the resolution of the LUT, i.e., the higher the resolution, the higher retrieval
accuracy. However, for multiple dimensional LUTs, the increase of resolution will increase the
table size exponentially, which will increase computational cost substantially for the table
building and inverse searching. Another possible method to increase the retrieval accuracy is
using different interpolation methods. For example, if the value of LUT varies non-linearly with
a variable, using high order interpolation method maybe better than using linear interpolation
method (Dannenberg, 1998).
**2.2.2 Method 2: Analytic transfer inverse model**
For a long time, various efforts have been devoted to the study of radiative transfer in the
atmosphere, including scattering, absorption, emission, and etc. (Chandrasekhar, 1960; Irvine
1964; Ivanov and Gutshabash 1974; van de Hulst, 1980, 2012; Ishimaru, 1999; Thomas and
Stamnes, 2002; Davis and Marshak, 2002; Kokhanovsky et al., 2003; Marshak and Davis, 2005;
Pandey et al., 2012; and etc.). In this study, we develop an analytic radiative transfer equation to
analyze the radiative transfer at oxygen A and B bands. Through solving the analytic equation,
we can retrieve the CTP information directly. The theory of CTP retrieval is similar for EPIC
oxygen A-band and B-band observation. Here we use oxygen A-band as an example to study the
radiative transfer model. For oxygen A-band, photon path length distribution is capable of
describing vital information related to a variety of cloud and atmospheric characteristics.
$$I_v(\mu, \varphi; \mu_0, \varphi_0) = I_0(\mu, \varphi; \mu_0, \varphi_0) \int_0^\infty p(l, \mu, \varphi; \mu_0, \varphi_0) e^{-\kappa_v l} dl \qquad (1)$$

Where, $p(l)$ is photon path length distribution, $\kappa_v$ is the gaseous absorption coefficient at wave
number $v$, $\mu = cos(\theta)$, $\mu_0 = cos(\theta_0)$, $(\theta, \varphi; \theta_0, \varphi_0)$ are zenith and azimuth angles for solar and
sensor view respectively, $I_0$ and $I_v$ are incident solar radiation and sensor measured solar radiation,
respectively.
When clouds exist, the incident solar radiation is reflected to TOA in three primary ways.
First, incident solar radiation is reflected by cloud top layer directly as a result of single
scattering. Second, the incident solar radiation will penetrate into the cloud and be reflected back
to TOA through cloud top via multiple scattering. Third, the incident solar radiation will pass
through the cloud and arrive at the surface, after that it is reflected back into the cloud and finally
scattered back to TOA through the cloud top. Due to the position of the EPIC instrument and the
long distance between EPIC and Earth, we can consider that solar zenith angle and sensor view
angle are nearly reverse. At oxygen A-band, the reflected solar radiation will be reduced due to
oxygen absorption depending on photon path length distributions. Absorption is negligible in
oxygen A-band's reference band. Oxygen A-band and its reference band are also attenuated by
airmass and aerosol through Rayleigh scattering and aerosol extinction. In the standard
atmospheric model, the optical depth of Rayleigh scattering ($\tau_{Ray}$) at oxygen A-band (B-band)
and its reference band is 0.026 (0.040) and 0.024 (0.042), respectively (Bodhaine et al., 1999).
The absolute difference of Rayleigh scattering optical depth ($\Delta\tau_{Ray} = \tau_{Ray}^{In-band} - \tau_{Ray}^{Ref}$) between
them is within 0.002. Compared to Rayleigh scattering, the difference of background aerosol
optical depth ($\Delta\tau_{Aer}$) between absorbing and reference bands is smaller, within 0.0005.
Therefore, the attenuations from Rayleigh scattering and aerosol extinction at EPIC oxygen
absorption and its reference band are close to each other. Thus, when we use the ratio of EPIC
measured reflectance at oxygen A-band and its reference band to derive the photon path length
distribution and retrieve cloud information such as CTP, the impact of Rayleigh scattering and
aerosol extinction can be simplified in the analytic transfer inverse model.
To simplify the analytic transfer inverse model for EPIC observations, we made a series
of assumptions, e.g., isotropic component, a plane-parallel homogenous cloud assumption with
quasi-Lambertian reflecting surfaces. These assumptions have been widely used in radiative
transfer calculation for cloud studies. In this model, $\mu$ and $\mu_0$ are the same as in Eq. (1), $\varphi$ is the
relative azimuth angle between Sun and satellite sensors; $A_{surf}$ is the surface albedo; $\tau_{O2}^{Top}$, $\tau_{O2}^{Base}$,
and $\tau_{O2}^{Surface}$ are oxygen A-band absorption optical depth from TOA to cloud top layer, cloud
bottom layer, and surface, respectively; $\Delta\tau_{O2}^{Above-Cld}$, $\Delta\tau_{O2}^{In-Cld}$ and $\Delta\tau_{O2}^{Below-Cld}$ are layered
oxygen A-band absorption optical depth above cloud, in cloud, and below-cloud, respectively;
functions $f$ mean their contribution to the ratio of measured reflectance at oxygen A-band ($R_A$)
and refrence band ($R_f$). The detailed analysis of EPIC analytic transfer inverse model is shown
as follows:
(1) **Above Cloud**: the reflected solar radiation is determined by the oxygen absorption optical
depth above the cloud and air mass directly.
$$f\big(\Delta\tau_{O2}^{Above-Cld}, \mu_0, \mu, \varphi\big) = f\big(\Delta\tau_{O2}^{Above-Cld}\big)f\big(\mu_0, \mu, \varphi\big)$$

$$= a_0 \tau_{O2}^{Top}\big(\frac{1}{\mu} + \frac{1}{\mu_0}\big) \tag{2}$$

Here, $a_0$ is a weight coefficient.
(2) **Within Cloud**: the reflected solar radiation is not only determined by oxygen absorption
optical depth above cloud and in-cloud, but also by penetration related factors, e.g., COD. Due to
photon penetration, oxygen parameter $\tau_{O2}^{Top}$ influences the enhanced path length absorption:
$$\Delta\tau_{O2}^{In-Cld} = \tau_{O2}^{Base} - \tau_{O2}^{Top} \tag{3}$$

Equivalence theorem (Irvine, 1964; Ivanov and Gutshabash, 1974; van de Hulst 1980) is used to
separate absorption from scattering:
$$f(\tau_{O2}^{Top}, \Delta\tau_{O2}^{In-Cld}, \mu_0, \mu, \varphi) = f(\tau_{O2}^{Top}, \Delta\tau_{O2}^{In-Cloud})f(\mu_0, \mu, \varphi)$$
$$= f(\tau_{O2}^{Top})f_1(\mu_0, \mu, \varphi) + f(\Delta\tau_{O2}^{In-Cloud})f_2(\mu_0, \mu, \varphi) \tag{4}$$
$f(\tau_{O2}^{Top})$ is determined by two absorption dependences: strong ($\sim \sqrt{\tau_{O2}^{Top}}$) and weak ($\sim \tau_{O2}^{Top}$).
$$f(\tau_{O2}^{Top}) = a_1\sqrt{\tau_{O2}^{Top}} + b_1(\tau_{O2}^{Top}) \tag{5}$$
Based on asymptotic approximation (*Kokhanovsky et al., 2003; Pandey et al., 2012*), the
reflection of a cloud without considering below cloud interaction is given by Eq. (6):
$$R(\tau, \mu, \mu_0, T) = R_0^\infty(\tau, \mu, \mu_0) - TK(\mu)K(\mu_0)$$
$$= R_0^\infty(\tau, f_1(\mu, \mu_0)) - Tf_2(\mu, \mu_0) \tag{6}$$
Here, $R_0^\infty$ is the reflectance of a semi-infinite cloud, $K(\mu)$ is the escape function of $\mu$, $T$ is global
transmittance of a cloud. $T$ can be estimated by Eq. (7), with the cloud optical thickness $\tau_{cld}$, the
asymmetry parameter , and a numerical constant $\alpha = 1.07$.
$$T = \frac{1}{0.75\tau_{cld}(1-g)+\alpha} \tag{7}$$
$f_1$ and $f_2$ functions have a quadratic form as follows:
$$f_{i-1} = a_i T + b_i(\mu + \mu_0) + c_i T(\mu + \mu_0) + d_i\mu\mu_0, i = 2,3 \tag{8}$$
Combining Eqs. (4), (5) and (8), we can get the Eq. (9):
$$f(\tau_{O2}^{Top}, \Delta t_{O2}^{Cld}, \mu_0, \mu, \varphi) = \left(a_1\sqrt{\tau_{O2}^{Top}} + b_1(\tau_{O2}^{Top})\right)\left(a_2 T + b_2(\mu + \mu_0) + c_2 T(\mu + \mu_0) + d_2\mu\mu_0\right)$$
$$+ \Delta\tau_{O2}^{In-Cloud}\left(a_3 T + b_3(\mu + \mu_0) + c_3 T(\mu + \mu_0) + d_3\mu\mu_0\right) \tag{9}$$

(3) **Below Cloud**: The equivalence theorem used for below cloud is similar to within cloud
(Kokhanovsky et al., 2003; Pandey et al., 2012).
$$f(\Delta\tau_{O2}^{Below-Cld}, \mu_0, \mu, \varphi) = T\,\tau_{O2}^{Surface}\frac{A_{Surf}}{1+(e_4*T+f_4)*A_{Surf}}$$
$$* (a_4 T + b_4(\mu + \mu_0) + c_4 T(\mu + \mu_0) + d_4\mu\mu_0) \tag{10}$$

Combining Eqs. (2), (9) and (10), we can get the total EPIC analytic transfer equation as
follows
$$-log\left(\frac{R_A}{R_f}\right) = f(\Delta\tau_{O2}^{Above-Cld}, \mu_0, \mu, \varphi) + f(\tau_{O2}^{Top}, \Delta\tau_{O2}^{Cld}, \mu_0, \mu, \varphi) +$$
$$f(\Delta\tau_{O2}^{Below-Cld}, \mu_0, \mu, \varphi) + \Delta\tau_{BG}\left(\frac{1}{\mu} + \frac{1}{\mu_0}\right) \tag{11}$$
In Eq. (11), $\Delta\tau_{BG}$ represents the sum of optical depth difference of background extinction (i.e.,
Rayleigh scattering $\Delta\tau_{Ray}$, aerosol extinction $\Delta\tau_{Aer}$, and O3 $\Delta\tau_{O3}$) between oxygen in-band and
reference band, as shown in Eq. (12).
$$\Delta\tau_{BG} = \Delta\tau_{Ray} + \Delta\tau_{Aer} + \Delta\tau_{O3} \qquad (12)$$

As stated in the previous subsection, in the standard atmospheric model with background aerosol
loading, ($\Delta\tau_{Ray}$, $\Delta\tau_{Aer}$, $\Delta\tau_{O3}$) is approximately (0.002, 0.0005, -0.0005) and (-0.002, -0.0005, -
0.002) respectively at oxygen A and B bands, thus $\Delta\tau_{BG}$ is approximately 0.002 and -0.0045
respectively at these two bands.
In this total analytic equation, there are 17 coefficients ($a_0, a_1, b_1, a_2, \ldots d_4, e_4, f_4$), which
can be calculated through nonlinear regression algorithm according to a series of simulated
values for different atmospheric conditions. Based on Eq. (11), we can finally obtain a quadratic
equation, $\mathbf{A}\sqrt{\tau_{O2}^{Top}}^2 + \mathbf{B}\sqrt{\tau_{O2}^{Top}} + \mathbf{C} = \mathbf{0}$, where the parameters A, B and C can be derived from
Eq. (11) directly, as shown in Eq. (13).
$$A = a_0\left(\frac{1}{\mu} + \frac{1}{\mu_0}\right) + b_1\left(a_2 T + b_2(\mu + \mu_0) + c_2 T(\mu + \mu_0) + d_2\mu\mu_0\right) \qquad (13.1)$$

$$B = a_1\left(a_2 T + b_2(\mu + \mu_0) + c_2 T(\mu + \mu_0) + d_2\mu\mu_0\right) \qquad (13.2)$$

$$C = -log\left(\frac{R_A}{R_f}\right) - \Delta\tau_{BG}\left(\frac{1}{\mu} + \frac{1}{\mu_0}\right) - \Delta\tau_{O2}^{In-Cloud}\left(a_3 T + b_3(\mu + \mu_0) + c_3 T(\mu + \mu_0) + d_3\mu\mu_0\right)$$

$$-T\,\tau_{O2}^{Surface}\frac{A_{Surf}}{1+(e_4*T+f_4)*A_{Surf}}\left(a_4 T + b_4(\mu + \mu_0) + c_4 T(\mu + \mu_0) + d_4\mu\mu_0\right) \quad (13.3)$$

When these parameters (i.e., A, B and C) are obtained from EPIC observation data and
other data source, we can easily solve the quadratic equation to retrieve cloud top O2 absorption
depth, and then CTP.

### 310 2.3 Detailed retrieval algorithm

As previously stated, in method 2, the analytic EPIC equation (i.e., Eq. (11)) is key for the
CTP retrieval. To derive the coefficients of Eq. (11), a series of model simulations for various
atmospheric conditions are needed. Thus, developing a radiative transfer model to simulate the
EPIC measurements at A- and B-bands and their reference bands is the first thing we need to
complete.

### 316 2.3.1 Oxygen A- and B-band absorption coefficients calculation

To simulate the EPIC measurements, one of the most important steps is calculating
oxygen absorption coefficients at oxygen A-band and B-band. In this step, the HITRAN 2016
database is used to provide the absorption parameters, and the LBLRTM package is used to
calculate oxygen absorption coefficients layer by layer. In our algorithm, the whole Earth
atmosphere is divided by 63 layers.
Since oxygen absorption coefficients are pressure (or pressure-squared) and temperature
dependent, and the line shapes ($k_i$) of oxygen A- and B-bands are well fitted as Lorentzian in the
lower atmosphere, the relationship can be written as follows:
$$k_i = \frac{S_i}{\pi}\frac{\alpha_i}{(v-v_i)^2+\alpha_i^2} \qquad (14)$$

$$\alpha_i = \alpha_i^0 \frac{P}{P_0} \left(\frac{T_0}{T}\right)^{\frac{1}{2}}, \; S_i = S(T_0)\frac{T_0}{T}\exp\left[1.439E\left(\frac{1}{T_0} - \frac{1}{T}\right)\right] \qquad (15)$$
Where $S_i$ is the line intensity, $v_i$ and $\alpha_i$ are the line center wave number and half width,
respectively; $P_0$ and $T_0$ are standard atmospheric pressure and temperature, respectively.
In the simulation of EPIC measurements, the atmospheric layer at a given layer-average
pressure can have drastically different temperature depending on the atmospheric profile in use.
To ensure the accuracy of simulation, we need to use the LBLRTM package to calculate oxygen
absorption coefficients for each pressure/temperature profile, which is a time-consuming process.
Our goal has been to find a simple and fast method to calculate oxygen absorption coefficients
for different atmospheric profiles. Based on the study of Chou and Kouvaris (1986), Min et al.
(2014) proposed a fast method to calculate oxygen absorption optical depth for any given
atmosphere by using a polynomial fitting function, as shown in Eq. (16).
$$\ln(A_{vLM}) = [a_0(v,P) + a_1(v,P) \times (T_{LM} - T_{mL}) + a_2(v,P) \times (T_{LM} - T_{mL})^2] \times \rho_{O_2} \quad (16)$$
Where $A_{vLM}$ is optical depths for layer L, spectral point v, and atmosphere model M; $\rho_{O_2}$ is
molecular column density ($\frac{molecules}{cm^2} \times 10^{-23}$); $T_{LM}$ is the average temperature for layer L for a
given atmosphere; and $T_{mL}$ is average temperature over all six typical geographic-seasonal model
atmospheres (M1 to M6, i.e., tropical model, mid-latitude summer model, mid-latitude winter
model, subarctic summer model, subarctic winter model, and the U.S. Standard (1976) model)
for layer L. To derive the coefficients $a_0$, $a_1$, and $a_2$, we first calculated oxygen optical depth
coefficients for all typical atmospheres (M1 to M6) by using LBLRTM package, and then
selected three of them (e.g., M1, M5, and M6) to calculate the polynomial fitting coefficients.
This method has been successfully used by Min et al. (2014) to simulate the high resolution
oxygen A-band measurements.
**2.3.2 Fast radiative transfer model for simulating high-resolution oxygen A- and B-bands**
At oxygen A and B absorption bands, there are lots of absorption lines, therefore we cannot
simply calculate narrowband mean optical depth and then calculate the radiation for various
atmospheric conditions when simulating EPIC narrowband measurements. The correct way is
described as follows: firstly, simulate the solar radiation spectrum $S(k(\lambda))$ under specific
atmospheric conditions, then integrate the spectrum with EPIC narrowband filter $R(k(\lambda))$ to
obtain simulated narrowband measurements (Eq. (17)).
$$R(\lambda) = \int S(k(\lambda))R(k(\lambda))d\lambda \neq R(\overline{k(\lambda)}) \qquad (17)$$
With the high spectrum resolution oxygen absorption coefficient data, we can simulate the
high resolution upward diffuse oxygen A-band or B-band spectrum through DISORT code
(Stamnes et al., 1988) for any given atmospheric condition, which has various surface albedo,
SZA, COD, CTH (CTP), and cloud geometric (pressure) thickness. However, due to the high
spectrum resolution, it is very time-consuming when performing line by line (LBL) calculations.
Thus, developing a fast radiative transfer model for simulating high resolution oxygen A-band
and B-band spectrum is necessary.
In this project, the double-$k$ approach is used to develop a fast radiative transfer model for
oxygen A-band and B-band respectively. [Min and Harrison 2004; Duan et al, 2005] proposed a
fast radiative transfer model. In their approach, the radiation from absorption and scattering
processes of cloud and aerosol are split into the single- and multiple-scattering components: The
single scattering component is computed line-by-line (LBL), while multiple scattering (second
order and higher) radiance is approximated.
$$I = I^{ss}(\lambda) + I^{ms}(\lambda)$$

$$\approx I^{ss}[Z^h(p,T), P^h, \lambda] + I^{ms}[Z^h(p,T), P^h, \lambda]$$

$$\approx I^{ss}[Z^h(p,T), P^h, \lambda] + I^{ms}[Z^l(p,T), P^l, \lambda]$$

$$\approx I^{ss}[Z^h(p,T), P^h, \lambda] + I^{ms}\{F[Z^l(p,T), P^l, k(\lambda_i)]\} \tag{18}$$

Eq. (18) is from Eq. (1) in Duan et al. (2005): *ss* and *ms* mean single and multiple scattering,
respectively. Z is the optical properties of the atmosphere as a function of pressure *p* and
temperature *T*, with P being the phase function of that layer. *h* and *l* represent higher and lower
number of layers and streams, respectively. F is the transform function between wave number
space and k space, defined from a finite set of $k(\lambda_i)$.

378       The application of Double-*k* approach in oxygen A-band has been presented in detail in
Duan et al. 2005. Here we take oxygen B-band as an example. The detailed fast radiative transfer
model for simulating high-resolution oxygen B-band is as follows: The first order scattering
radiance is calculated accurately by using a higher number of layers and streams for all required
wavenumber grid points. The multiple-scattering component is extrapolated and/or interpolated
from a finite set of calculations in the space of two integrated gaseous absorption optical depths
to the wavenumber grids: a double-*k* approach. The double-*k* approach substantially reduces the
error due to the uncorrelated nature of overlapping absorption lines. More importantly, these
finite multiple-scattering radiances at specific *k* values are computed with a reduced number of
layers and/or streams in the forward radiative transfer model. To simulate an oxygen B-band
spectrum with high accuracy, 33 *k* values and 99 calculations of radiative transfer are chosen in
our program. This results in around a hundred-fold time reduction with respect to the standard
forward radiative transfer calculation.

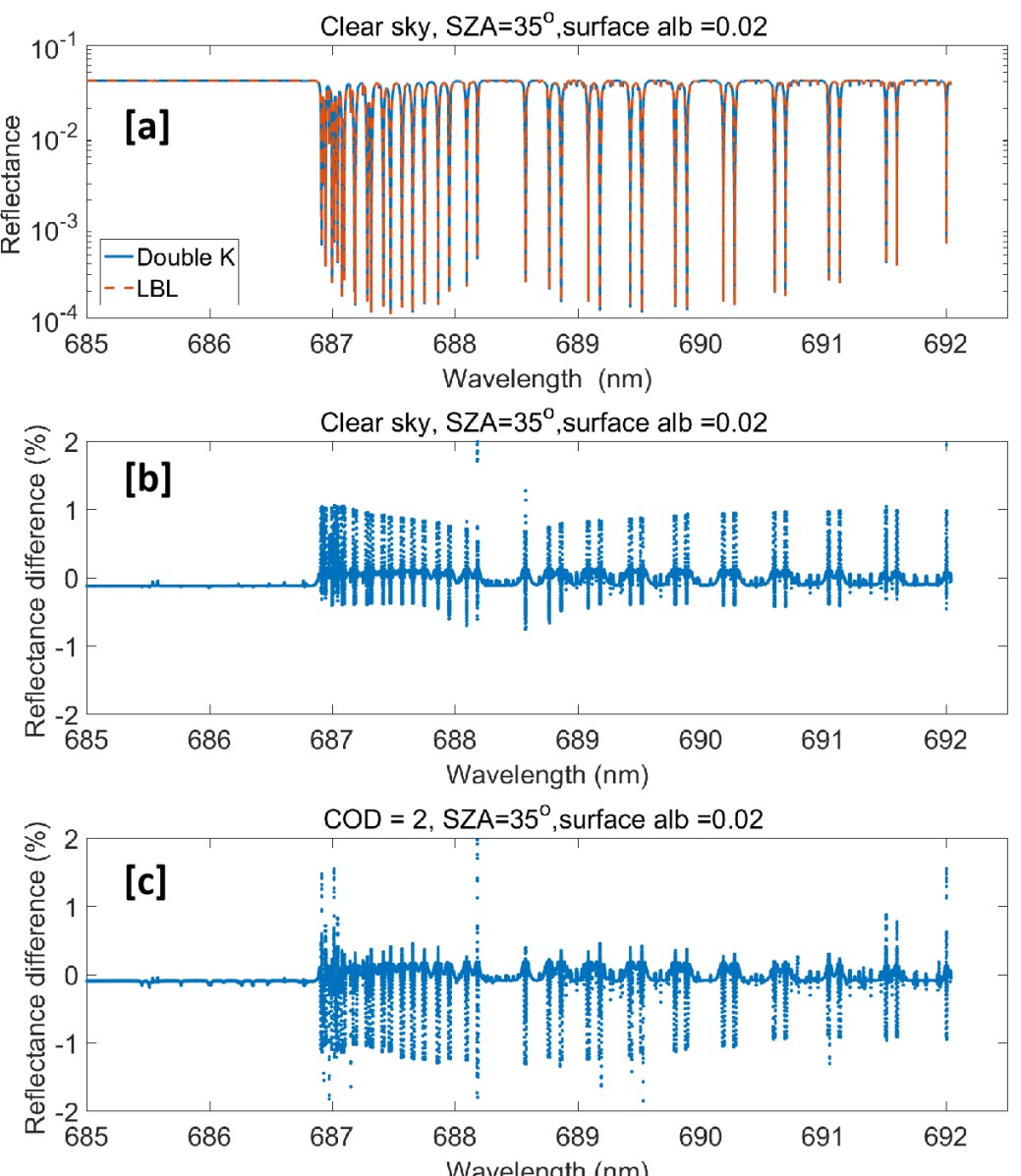

**Figure 2**. (a) High resolution reflectance at EPIC O2 B-Band simulated by fast radiative model (double-k) and benchmark (LBL); Difference between simulated reflectance by (b) double-k and LBL for a clear sky case and (c) a thin liquid water cloud case with COD=2. Here SZA and view angle =35°, surface albedo = 0.02, aerosol optical depth = 0.08, and reflectance difference (%) = 100*((double-k) – LBL)/LBL.

As shown in Fig. 2, under clear sky and thin liquid water cloud situations, the simulated high resolution upward diffuse oxygen B-band spectra from LBL calculation and double-k approach are compared. The spectrum difference between LBL calculation and double-k approach is very small (Fig. 2a). Under both situations, most of the relative difference between these two methods are under 0.5%. The obvious relative difference (>1%) occurs only in the wavelength range with high absorption optical depth, which has little contribution to the integrated solar radiation. Therefore, for the simulated narrowband measurements at EPIC

oxygen B-band, the relative difference between LBL and double-k approach is much smaller than that of the high resolution spectrum, which is less than 0.1% for clear day. Compared to clear sky situation, the relative difference for cloud situations can be bigger. As shown in Table 1, the relative difference is -0.06% and -0.32% for typical high level optical thin cloud and low-level thick cloud situations, respectively. The comparison of simulated narrowband measurement at EPIC oxygen A-band channel (764 nm) is also shown in Table 1, the relative differences between LBL and double-k approach are -0.06%, 0.21% and 0.23% for clear day, high level thin cloud and low level thick cloud cases, respectively. In general, the accuracy of double-k approach for both oxygen A and B absorption bands is high.

**Table 1.** Comparison of simulated narrowband measurement at EPIC A- and B-Band channels

| Case (SZA=35, surface albedo =0.02) | | Line by Line | Double k | Relative Difference |
|---|---|---|---|---|
| Clear Day | 688 nm | 0.026963 | 0.026985 | +0.08% |
| | 764 nm | 0.013979 | 0.013970 | -0.06% |
| Thin cloud (COD=2, 8.3-8.5 km, liquid) | 688 nm | 0.098444 | 0.098131 | -0.32% |
| | 764 nm | 0.071359 | 0.071507 | +0.21% |
| Thick cloud (COD=16, 1.5-2.9 km, liquid) | 688 nm | 0.396354 | 0.396117 | -0.06% |
| | 764 nm | 0.233937 | 0.234485 | +0.23% |

### 2.3.3 Simulation of oxygen A- and B-bands for different atmospheric conditions

Using the EPIC measurement simulation package, we made a series of simulations with different settings for surface albedo, solar zenith angle, COD, CTH (CTP), and cloud geometric (pressure) thickness (or cloud bottom height). The results of these simulations consist of a data table, which can be used not only to calculate the coefficients for the analytic equation, but also to study the sensitivity of every variable.

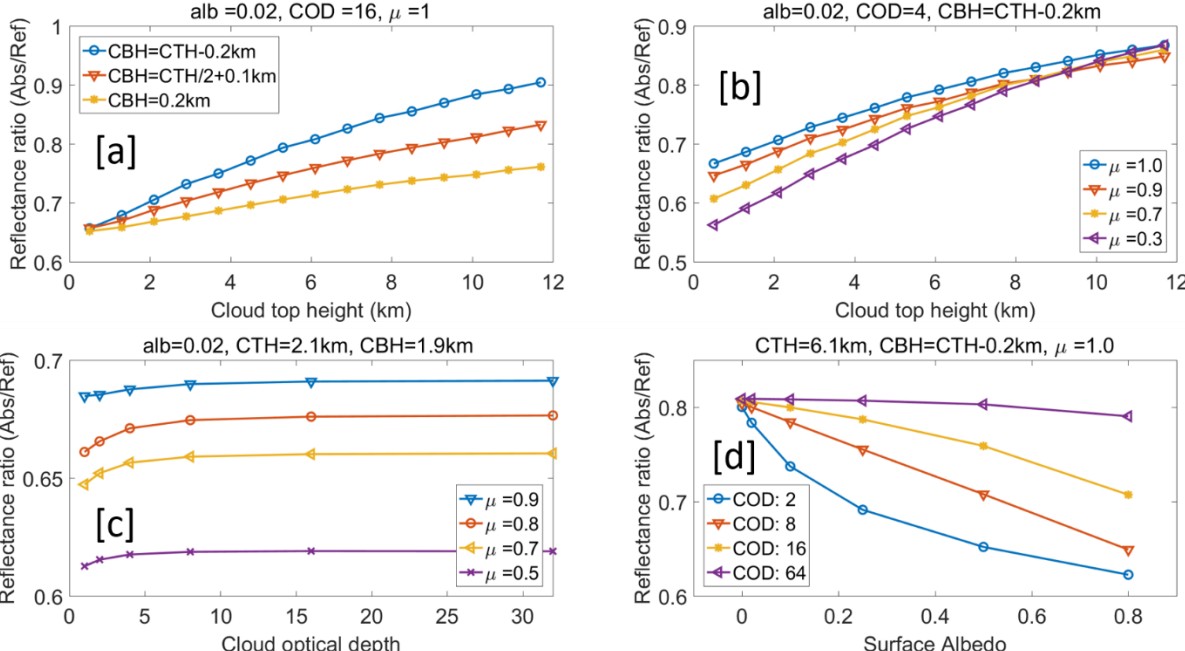

**Figure 3.** Ratio of simulated reflectance measurements for EPIC B-band to B-band reference with different surface albedo (alb), COD, $\mu$ (cosine of solar zenith angle), cloud top height (CTH) and cloud bottom height (CBH).

According to the previous theory study, the ratio of reflectance radiance (i.e., absorption to the reference) at TOA is determined by the photon path length distribution at oxygen A/B bands: the larger the mean photon path length, the stronger the absorption, and the smaller the reflectance ratio. To make the figures easy to view and understand, we use cloud top and bottom geometric height to represent CTP and thickness information in Fig. 3. As shown in Fig. 3a, the ratio of upward diffuse radiance at oxygen B-band and its reference band is sensitive to the cloud top height (pressure). The higher the CTH, the larger the ratio. At the same time, this ratio is affected by the cloud bottom height (or cloud geometric thickness) when the other cloud parameters are fixed, the lower the cloud bottom (or the larger the cloud geometric thickness), the smaller the ratio. It is consistent with the theory analysis: (1) the higher the CTH, the shorter the mean photon path length, and the weaker the absorption; (2) when the COD is given, larger cloud geometric thickness means smaller cloud density, then the sunlight can penetrate deeper into the cloud, which results in a longer mean photon path length. In Fig. 3b, for clouds with given CTH, COD and geometric thickness, the ratio decreases with the solar and view angles. This can be understood as: the larger the solar and view angles, the longer the mean photon pathlength, and the stronger the absorption. In Fig. 3c, for clouds with given CTH and geometric thickness, when the COD is small (e.g., COD <5), the reflectance ratio increases with COD. However, when COD is larger than 16, the effect of COD is small. This is because the larger the COD, the shallower the sunlight penetration, and the shorter the mean photon pathlength. In Fig. 3d, for clouds with given COD, CTP, and geometric thickness, the ratio decreases with surface albedo. The smaller the COD, the stronger the impact of the surface albedo. This is because the thick cloud prevents the incident sunlight from passing through it to reach the surface, and also prevents the reflected light from going back to the TOA.

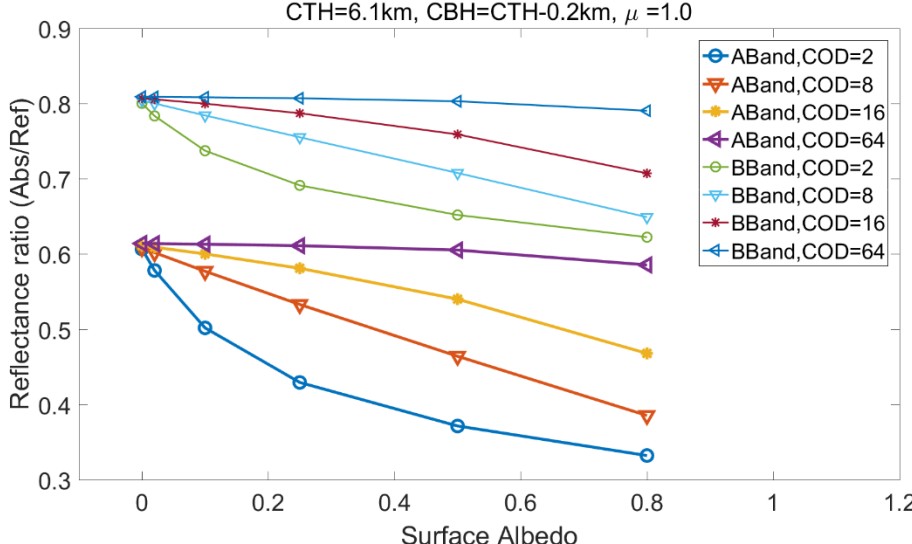

449

Figure 4. Ratio of simulated reflectance measurements for EPIC A and B absorption band to reference band with different surface albedo.

For oxygen A-band, the ratio of upward diffuse at absorption and reference bands shows similar characteristics as for oxygen B-band. Compared to oxygen B-band, under the same atmospheric conditions, the oxygen absorption at A-band is stronger, and the ratio of A-band to its reference band has smaller values (shown in Fig. 4). As stated previously, for land area that covered with plants, the surface albedo may change substantially from oxygen B-band to A-band due to the presence of the red edge. Therefore, accurate spectral data of surface albedo for CTP retrieval is vitally important, especially for optically thin clouds.

## 3. Application and validation of the CTP retrieval method

### 3.1 Case studies of CTP retrieval

The dataset of DSCOVR EPIC measurements at GMT 00:17:51 on July 25, 2016 is used for the case studies. The reflectance at oxygen A and B bands with related solar zenith and viewing angles are obtained from the EPIC level 1B data; COD information (retrieved from other EPIC channels) is obtained from EPIC level 2 data. The surface albedo data is obtained from Global Ozone Monitoring Experiment 2 (GOME-2) Surface Lambertian-equivalent reflectivity (LER) data. The detailed information of dataset is shown in the acknowledgements and dataset. To reduce the impact of the Earth surface, we selected the region located in spatial range of (75° S to 85° N, 177° W to 175° W) for case studies, which is mainly covered by ocean. To constrain the influence of surface albedo and broken clouds, only pixels with total cloud cover (i.e., EPIC Cloud mask = 4), surface albedo less than 0.05, and liquid assumed COD larger than 3 are considered. In the selected region, around 10000 pixels are finally chosen for case studies.

In our retrieval algorithm, we have two kinds of retrieval results: baseline CTP and retrieved CTP. The baseline CTP is used as a reference for the retrieved CTP. It is similar to the effective CTP in Yang et al., (2019), which does not consider cloud penetration. The retrieved CTP is calculated by the analytic equation, which considers the in-cloud and below-cloud interaction.

During the baseline CTP calculation, the impact of penetration in-cloud is ignored, and the incident light that reached cloud top is assumed reflected back directly. As shown in Eq. (19), the baseline absorption optical depth $\tau_{base}$ is derived from the ratio of upward diffuse at absorption bands and their reference bands directly. According to the model calculated oxygen A and B bands absorption optical depth profile at the specific solar zenith angle, the baseline CTP can be derived directly.

$$\tau_{base} = \log\left(-\frac{R_{abs}}{R_{ref}}\right)/(\frac{1}{\cos(\theta_{sza})} + \frac{1}{\cos(\theta_{view})}) \tag{19}$$

As shown in Fig. 5, the baseline CTP value at A-band is slightly higher than the effective CTP from NASA ASDC L2 data. But the baseline CTP value at B-band is substantially higher than the effective CTP from NASA ASDC L2 data. For both A-band and B-band, the difference between baseline CTP and effective CTP increases with the CTP. For low-level clouds, the mean differences of them are up to 60 mb and 100 mb at A-band and B-band, respectively. The difference may be mainly from the calculation of oxygen A and B bands absorption coefficients or the absorption optical depth profile.

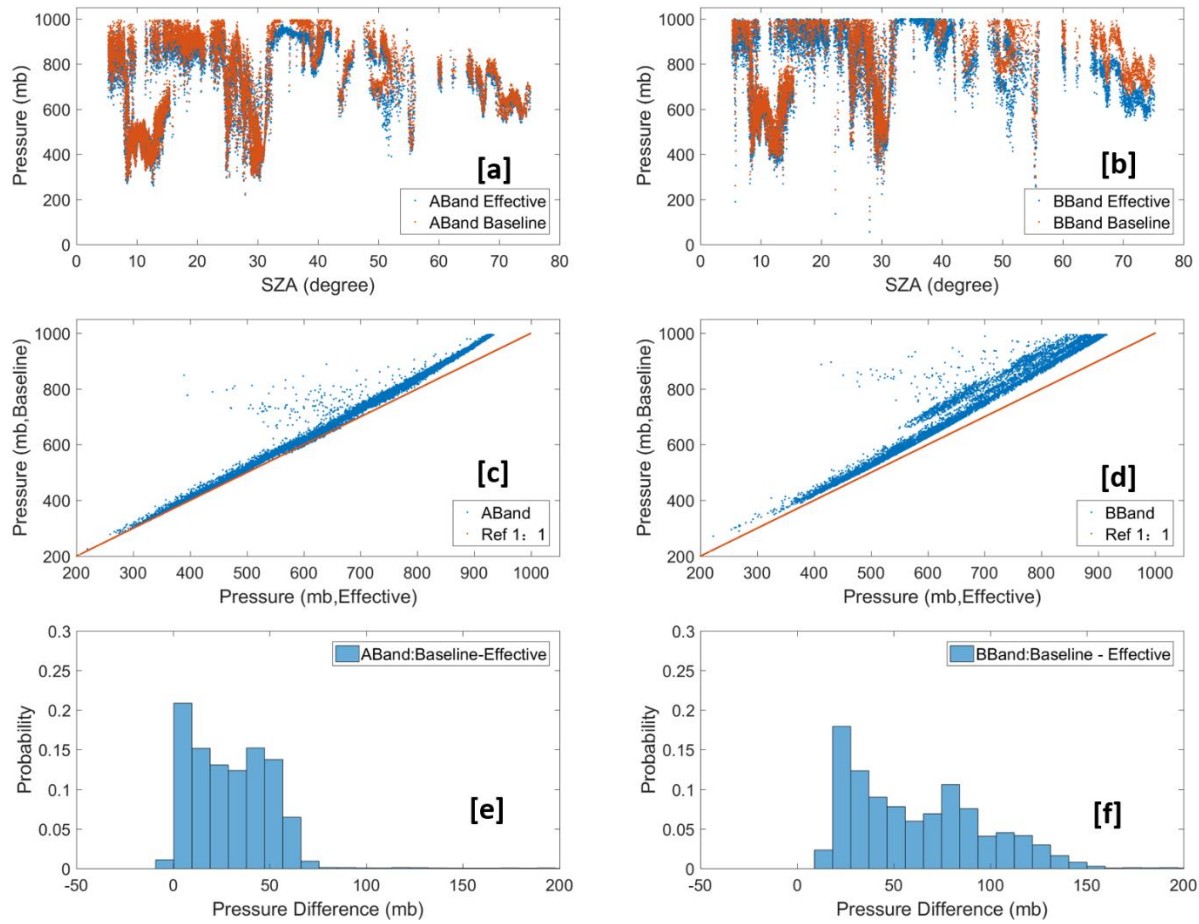

**Figure 5.** The comparison of effective CTPs (reference from NASA ASDC data) and baseline CTPs from our retrieval algorithm for EPIC A and B bands.

Based on the simulated reflectance ratio under different atmospheric conditions, we can
calculate the coefficients for the analytic radiative transfer equations by using a nonlinear fitting
algorithm. The coefficients for different SZA's are calculated individually to reduce the fitting
error. Based on the calculated coefficients, we can retrieve the CTP with DISCOVR EPIC
observation data at oxygen A and B bands.
During the CTP retrieval, with the exception of the previously mentioned analytic
equation coefficients, we can get the surface albedo data from GOME, obtain reflectance data,
solar zenith and view angles, COD, etc. from the NASA ASDC data file. Another very important
step in the retrieval processing is the acquisition of cloud pressure thickness data, which has a
substantial impact on the retrieval results. We currently use a statistical approach (i.e., cloud
pressure thickness (mb) = 2.5* COD +23) to estimate the cloud pressure thickness based on
COD.  As shown in Figs. 6a-6d, the retrieved CTP when considering cloud penetration is smaller
than baseline CTP. For this case, the mean difference between baseline CTP and retrieved CTP
for oxygen A-band and B-bands are around 57 mb and 85 mb, respectively, which is consistent
with theoretical expectations. For clouds with a given CTP, the mean photon path length will
increase substantially when considering cloud penetration. A decrease in retrieved CTP will
result in order to match the measurement ratio of absorption to reference. Compared to the O2 A-
band, both baseline CTP and retrieved CTP for the O2 B-band are larger (Figs. 6e-6h). This is
because the absorption of solar radiation in the O2 B-band is weaker than that of the O2 A-band,
and the incident light at oxygen B-band can penetrate deeper into the cloud, allowing more light
to pass through. The difference in retrieved CTP between B band and A band (approx. 93 mb
with standard deviation of 83 mb) is generally reduced in comparison to baseline B band and A
band (approx. 114 mb with standard deviation of 73 mb). This indicates, as expected, more
photon penetration correction for B-band than A-band.

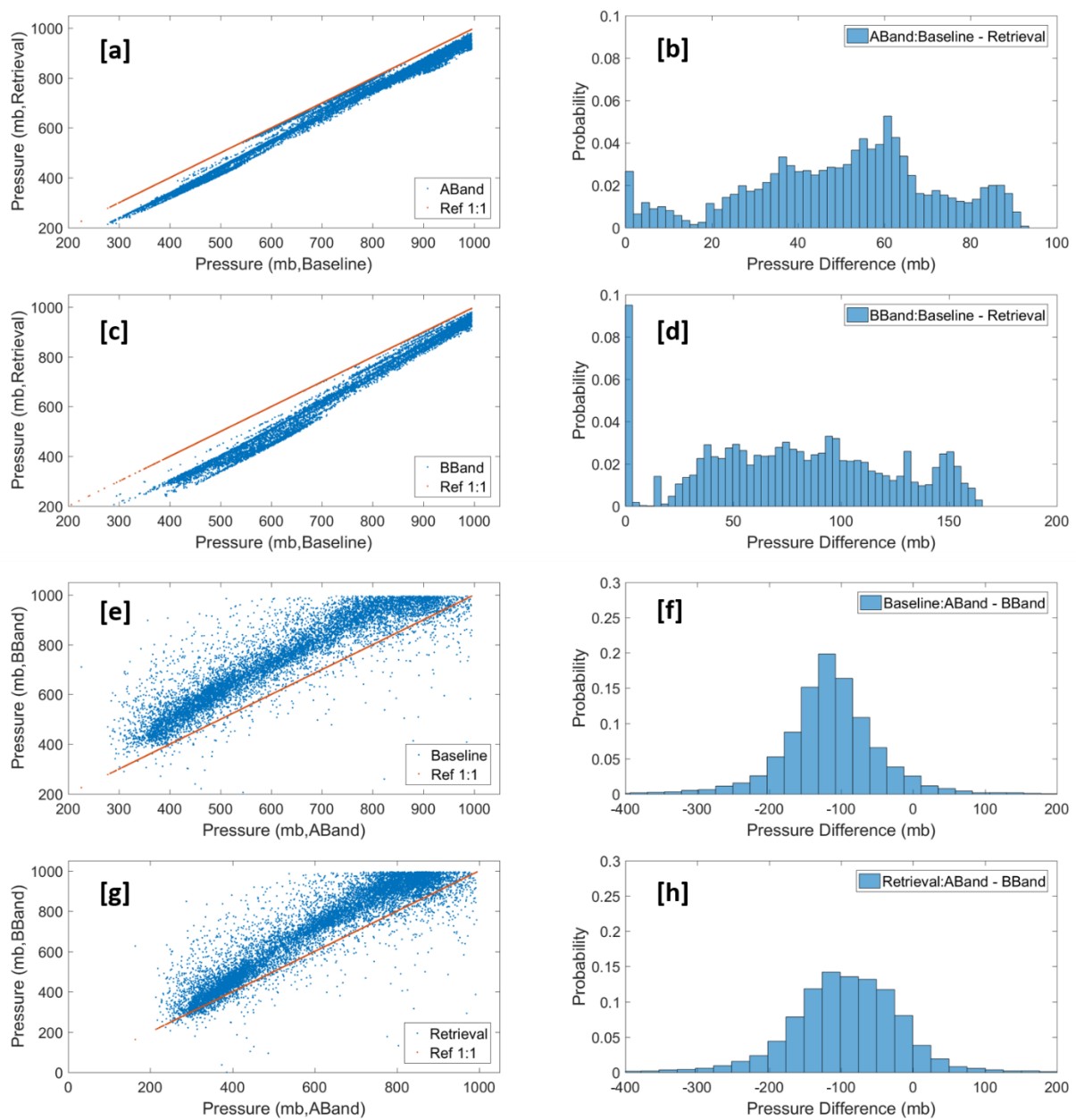


**Figure 6.** (a-d) The comparison of retrieved CTPs and baseline CTPs for EPIC A and B bands; (e-f) the comparison of retrieved CTPs and baseline CTPs between EPIC A- and B- bands.

We also used the LUT based method to do the retrieval for the same observation data, because both methods share the same EPIC simulation package and the same simulated data table, the results of which are similar.

## 3.2 Validation of the retrieval method

To validate the analytic transfer inverse model method for CTP retrieval, we used another independent measurement of CTP, i.e., cloud layer top pressure from Cloud-Aerosol Lidar and Infrared Pathfinder Satellite Observations (CALIPSO, Vaughan et al., 2014) as a reference. For

the previously stated case, i.e., DSCOVR EPIC measurements at GMT 00:17:51 on July 25,
2016, we used the cloud layer data from CALIPSO IIR Version 4.2 Level 2 product with 5 km
resolution at GMT 00:01:47 on July 25, 2016 as its reference to do validation. To constrain the
error from spatial differences between different satellite measurements, we only chose the pixels
of EPIC and CALIPSO measurements with a spatial distance of within 0.1° (degree of latitude or
longitude) to make comparisons. For the EPIC measurements, the same as previously stated,
only pixels with total cloud cover (i.e., EPIC Cloud mask = 4), surface albedo less than 0.05, and
liquid assumed COD larger than 3 are considered.  As shown in Fig. 7a, there are a series of
pixels (around 400 cases) from EPIC and CALISPO measurements can be used for the validation
analysis. For the convenience of reading, we perform the analyses by using the case number as x
axis.  Fig. 7b shows the comparisons of cloud layer top pressure from CALIPSO and different
CTPs (i.e., effective CTP, baseline CTP, and retrieved CTP) from EPIC measurements. Fig. 7c
shows the cloud layer number measured by CALIPSO. According to Figs. 7b and 7c, we can get
some results: under single layer cloud situations, the CTPs derived from EPIC measurements are
close to the CTP from CALISPO; under multi-layer cloud situations, the CTP derived from EPIC
measurements are larger than the CTP from CALISPO. Fig. 7d shows the expanded view of the
Fig. 7b for some cases under single layer cloud situations. For these single layer cloud cases
(with case number 46 ~ 156), the mean values of CTP of CALIPSO, EPIC effective, EPIC
baseline and EPIC retrieval are 846, 834, 866 and 850 mb, respectively. Compared to the CTP
from CALIPSO measurements , the EPIC effective and  baseline CTPs are 12 mb smaller or 20
mb larger, respectively; the EPIC retrieval with consideration of photon penetration is only 4 mb
larger. This shows that our method for the CTP retrieval is valid and accurate under single layer
cloud situations with COD > 3 and low surface albedo.  Under multi-level cloud situations, the
high-level clouds are often thin clouds, which can be detected by CALIPSO but hard to derive by
our retrieval method. It is because the EPIC retrieved CTP mainly shows the pressure of cloud
layer that reflects the major part of incident sun light.

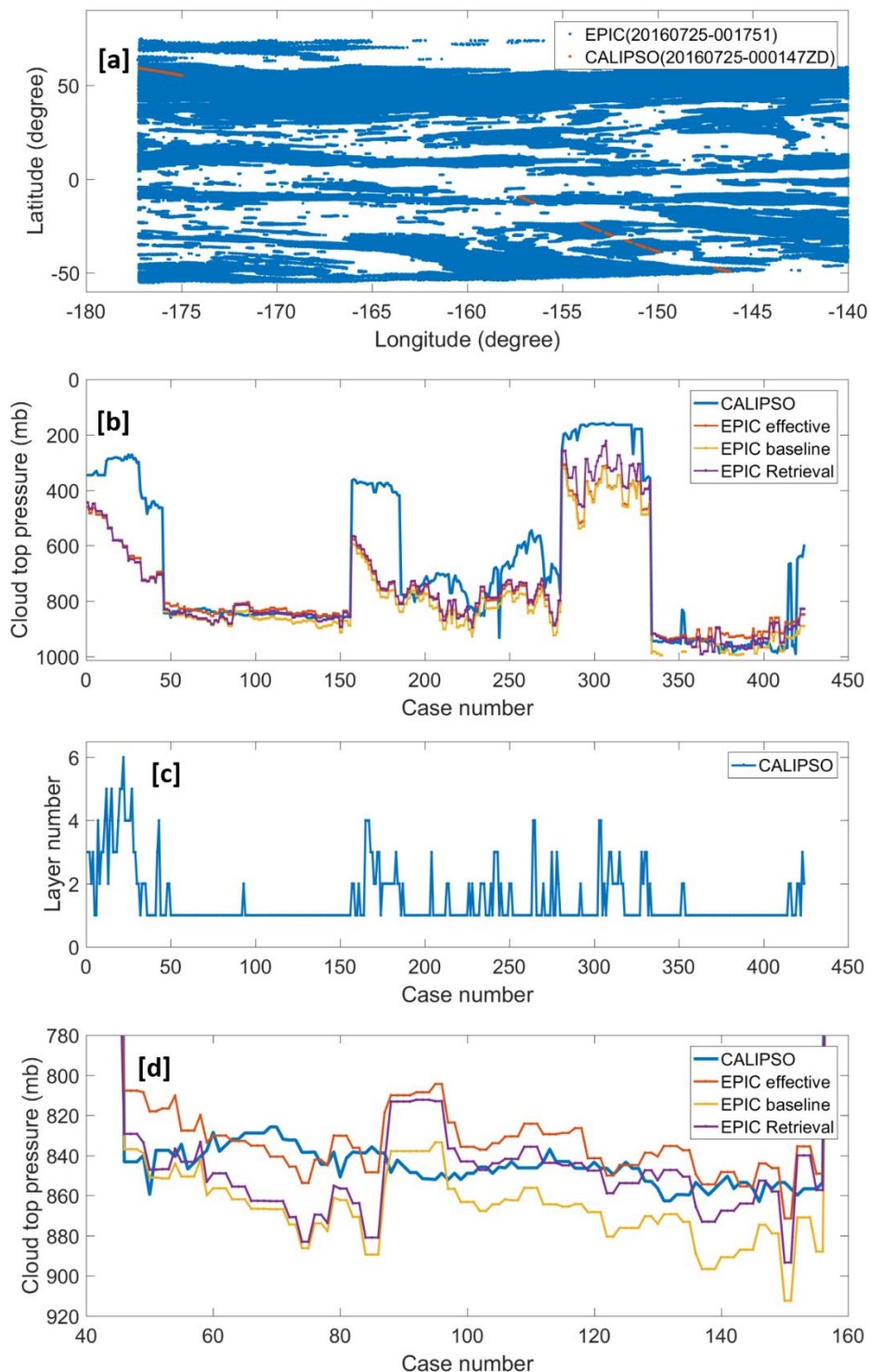


Figure 7. (a) The geolocation match of EPIC measurement at GMT 00:17:51 and CALISPO
measurement at GMT 00:01:47 on July 25, 2016; (b) the comparisons of cloud layer top pressure
from CALIPSO measurements and the CTPs derived from EPIC measurements; (c) the cloud
layer number from CALIPSO measurements; and (d) the expanded view of (b) for some cases
under single layer cloud situations.

564

## 3.3 Retrieval of global observation

We applied our retrieval algorithm on the global DISCOVR EPIC measurement data at oxygen A and B bands. During the retrieval, only pixels with total cloud cover (i.e., cloud mask index of 4), surface albedo < 0.25, and COD >= 3 are considered. To make the pictures easy to visualize and analyze all invalid values are plot as white (or blank) pixels.

Figure 8a shows the synthesized RGB picture of EPIC measurements at GMT time 00:17:51 on July 25, 2016. At this point in time the sun light covers most of the Pacific Ocean. In this figure, the white pixels represent cloud cover. Figure 8b shows the global COD (NASA ASDC L2 data), in which the white areas and colorful areas indicate the clear sky areas and cloudy areas, respectively. On the whole, the cloudy areas are consistent with the RGB image. The highlight (red) areas indicate that the cloud systems there contain optically heavy clouds. Figure 8c shows the A-band effective CTP (NASA ASDC L2 data), where the white areas indicate clear sky or no valid values, warm (brown) and cold (blue) color areas indicate high-level and low-level clouds, respectively. According to the A-band effective CTP, the high-level clouds are dominant in the equatorial area, and the low-level clouds play a major role in the cloud systems in the Northern Pacific area. Figures 8d and 8e show the baseline and retrieved CTP at A-band, respectively, which cloudy areas are consistent with the A-band effective CTP image on the whole. Due to the filtering setting in the CTP retrieval algorithm, there are more white pixels (invalid values) in these two figures. The difference of A-band retrieved CTP and A-band effective CTP is shown in Fig. 8d. The A-band retrieved CTP is overall smaller than A-band effective CTP, which difference is within 100 mb. The highlighted (brown or red) areas are located in the high level clouds areas or large COD areas. This indicates that the complexity of cloud system has significant impact on the CTP retrieval. Figures 8g and 8h show the baseline and retrieved CTP in B-band respectively, which are similar to, but greater than the A-band. As shown in Fig. 8i, the retrieved CTP at EPIC B-band is overall significantly larger than the retrieved CTP at EPIC A-band, which mean difference is up to 200 mb.

591

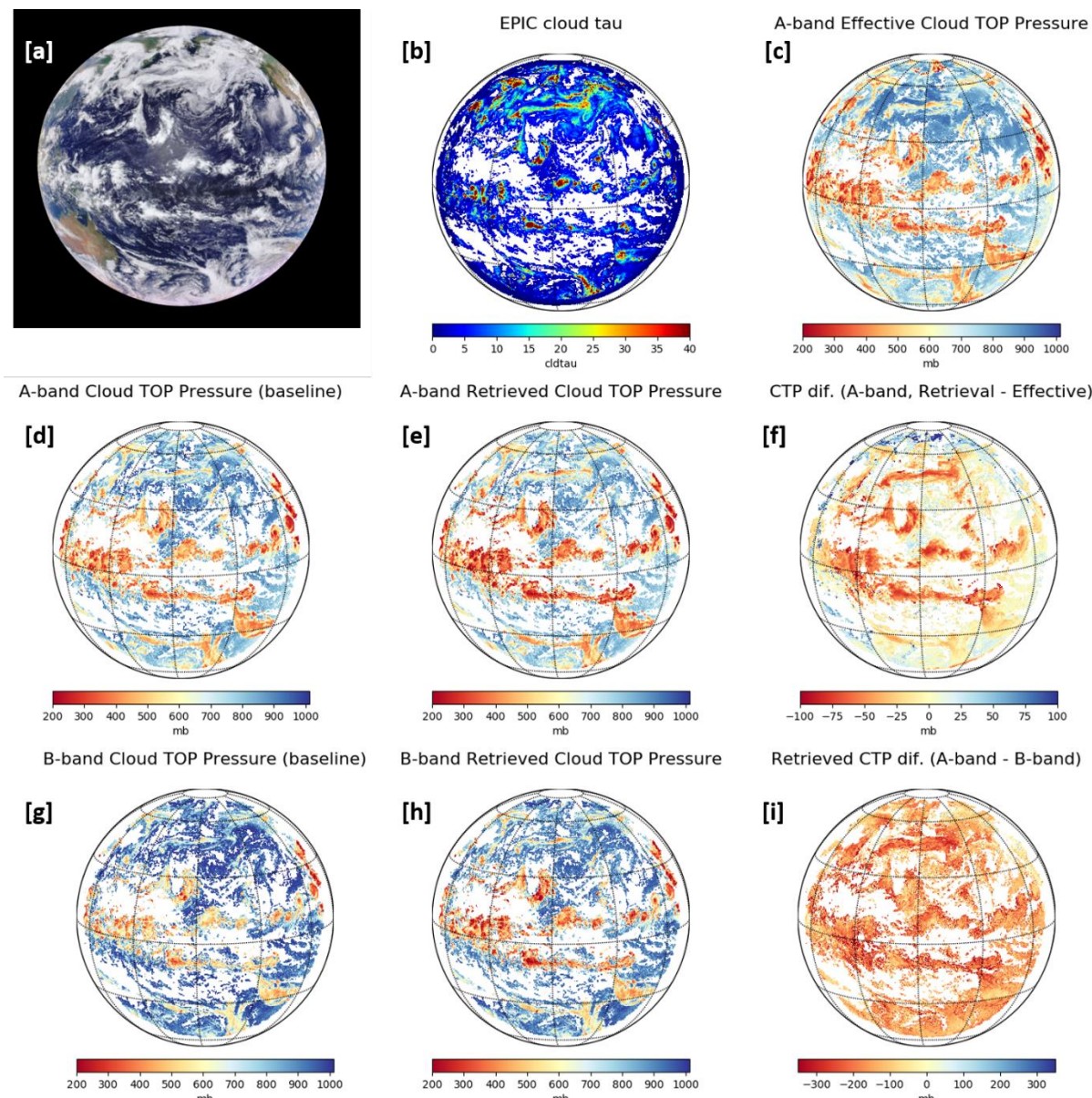

**Figure 8.** (a) RGB image from DSCOVR EPIC measurement at GMT time 00:17:51 on July 25, 2016; (b) and (c) COD (liquid assumption) and A-band effective CTP from NASA ASDC EPIC L2 products; (d) and (e) Baseline and retrieved CTP derived from EPIC A-band measurement; (f) the difference of A-band retrieved CTP and A-band effective CTP; (g) and (h) Baseline and retrieved CTP derived from EPIC B-band measurement; and (i) the difference of retrieved CTP between EPIC A-band and B-band.

As previously stated in Sect. 3.2: under single-layer cloud situations, the CTPs derived from EPIC A-band measurements have good agreement with the CTP from CALIPSO measurements; under multiple-layer cloud situations, the CTPs derived from EPIC measurements may be larger than the CTPs of high level thin-clouds due to the effect of photon penetration. Therefore, in the global range, for the large scale low-level stratus clouds, the retrieved CTPs from EPIC A-band measurements should agree well with the actual value of CTPs, but for the complex cloud system

with multiple-layer clouds, the CTPs derived from EPIC A-band measurements may be larger
than that of high level thin-clouds.

## 4. Conclusion

The in-cloud photon penetration has significant impacts on the CTP retrieval when using
DSCOVR EPIC oxygen A- and B- band measurements.  To address this issue, we proposed two
methods, (1) the LUT based method and (2) the analytic transfer inverse model method for CTP
retrieval with consideration of in-cloud photon penetration. In the analytic transfer inverse model
method, we build an analytic equation that represents the reflection at TOA from above cloud,
in-cloud, and below-cloud, respectively.  The coefficients of this analytic equation can be
derived from a series of EPIC simulations under different atmospheric conditions using a non-
linear regression algorithm. With EPIC observation data, the related solar zenith and sensor view
angle, surface albedo data, COD, and estimated cloud pressure thickness, we can retrieve the
CTP by solving the analytic equation.
We developed a package for the DSCOVR EPIC measurement simulation. The high
resolution radiation spectrum must be simulated first and then integrated with the EPIC filter
function in order to accurately simulate EPIC measurements. Because this process is highly time-
consuming, a polynomial fitting function is used when calculating the oxygen absorption
coefficients under different atmospheric conditions. At the same time, the double-k approach is
applied to do the high-resolution spectrum simulation to further reduce time-costs, which can
obtain high accuracy results with hundred-fold time reduction. The results of the EPIC
simulation measurements are consistent with theoretical analysis.
Based on the EPIC simulation measurements, we derived a series of coefficients from
various solar zenith angles for the analytic EPIC equations. Using these coefficients, we
performed CTP retrieval for real EPIC observation data. We have two kinds of retrieval results:
baseline CTP and retrieved CTP. The baseline CTP is similar to the effective CTP in Yang et al.,
(2019), which does not consider cloud penetration. The retrieved CTP is derived by solving the
analytic equation, with consideration of the in-cloud and below-cloud interactions. Compared to
the effective CTP provided by NASA ASDC L2 data, the baseline CTP value at A-band is
slightly higher, but the baseline CTP value at B-band is substantially higher. The retrieved CTP
for both oxygen A- and B- bands is smaller than the related baseline CTP. At the same time,
compared to the oxygen A-band, both baseline CTP and retrieved CTP at oxygen B-band is
larger. The cloud layer top pressure from CALIPSO measurements is used to validate the CTP
derived from EPIC measurement. Under single-layer cloud situations, the retrieved CTPs for
oxygen A-band agree well with the CTPs from CALIPSO, which mean difference is within 5 mb
in the case study. Under multiple-layer cloud situations, the CTPs derived from EPIC
measurements may be larger than the CTPs of high level thin-clouds due to the effect of photon
penetration.
Currently, this analytical transfer model method can only retrieve CTP, and it still need
cloud pressure thickness as an input parameter. However, in the satellite observations, both CTP
and cloud pressure thickness are unknown. The estimation or assumption of cloud pressure
thickness will bring in extra error in CTP retrieval. In the near future, we plan to address this
issue.

**Data availability**

Dataset of DSCOVR EPIC Level 1B can be found in https://eosweb.larc.nasa.gov/project/dscovr/ dscovr_epic_l1b_2; dataset of EPIC Level 2 can be found in https://eosweb.larc.nasa.gov/ project/dscovr/dscovr_epic_l2_cloud_01; dataset of surface albedo from GOME can be found in http://temis.nl/surface/gome2_ler/databases/; dataset of cloud layer data from CALIPSO can be found in https://eosweb.larc.nasa.gov/project/calipso/cal_lid_l2_05kmclay_standard_v4_20.

**Author contributions**.

All authors contributed to planning and writing of the paper. QM initiated and led the EPIC CTP retrieval project, designed the analytic transfer inverse model for EPIC observation. BY developed a fast radiative transfer model for simulating high resolution oxygen B-band, implemented the CTP retrieval algorithms by using the analytic transfer inverse model and look-up table method, and drafted the paper. EM calculated the high resolution Oxygen absorption coefficients by using the LBLRTM model and HITRAN database. YY provided access to the EPIC Level 1B and Level 2 products and provided guidance for the evaluation of EPIC CTP retrieval. AM and ABD conducted the EPIC related studies and provided guidance for the design of EPIC CTP retrieval algorithm with the consideration of in-cloud photon penetration.

**Competing interests**

The authors declare that they have no conflict of interest.

**Acknowledgements**

This work was supported partially by NASA's Research Opportunities in Space and Earth Science (ROSES) program element for DSCOVR Earth Science Algorithms managed by Dr. Richard Eckman, by the National Science Foundation (NSF) under contract AGS-1608735; and by the National Oceanic and Atmospheric Administration (NOAA) Educational Partnership Program with Minority Serving Institutions cooperative agreement #NA11SEC4810003.

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
