# Peer review of "Cloud top pressure retrieval with DSCOVR-EPIC oxygen A and B bands observation"

_Atmospheric Measurement Techniques, 2019_

## Referee Comment (RC1) · Anonymous Referee #1 · 25 Mar 2020

This paper introduces an algorithm for the determination of the cloud top pressure inferred from measurements of oxygen absorption in the NIR by the EPIC sensor on-board the DSCVR platform. The topic is important and appropriate for the journal.

The paper shows some sound science and the authors have structured their manuscript in the correct way. All major sections needed to present a retrieval algorithm are, in my opinion, addressed. However, major improvements are still needed and I will be willing to evaluate a revised version of the paper. I bullet-list improvements and remarks "general comments" section and then I delve in the explanation of specifics later on.

*** General main comments

[Figure]

- Before any scientific content scrutiny, I suggest to throughly check punctuation, syntax and word C-(c)-apitalization and arrangement prior publication. Copernicus service should definitely help here, but also and foremost checks by the english native co-authors. Uneven sentences or awkward wording are present throughout the manuscript and are too many to be listed by a referee. This will help to showcase the logic of the method and the importance of the results.

- In the introduction state clearly and make explicit the difference with Yang et al. JQSRT, 2013. As both papers share the same goal, data source and co-authors, it is important to highlight the advancement achieved in this paper with respect to previous literature. Some scientific insights about the difference between the A and the B-band are given in Yang et al. but are put to little of any use in this work. One would expect some science advancement and not a mere application or repetition of a method. All my criticism and required improvements naturally follow from this remark.

- Therefore, the treatment of aerosols is overly simplified or neglected together with error analysis as function of cloud optical thickness or cloud cover, since we know that from the remote sensing perspective these two quantities are connected.

- The coefficients A, B, C (P6 L199) must be presented otherwise the reader is not equipped with the knowledge to replicate the results.

- The presentation and analysis of the results is suboptimal. Without proper and customary validation with external independent data sets little knowledge can be won about the applicability ranges of the presented method in real geophysical scenarios, which is one of the stated goals of the paper, otherwise Section 5 would not be presented.

*** Specific comments to individual sections

- Abstract

P1 L28: why "obviously"? It is not a straightforward inference and it is not objective, but

subjective instead. Please, remove it from the abstract.

P1 L29-30: could you provide quantitative figures for the comparisons? Something like "Out of N cases, we found an average bias between CTP b- and A-band of xxx hPa $\pm$ xxx hPa_stdv".

P2 L44: "their atmospheric profiles"? You may want to check this, because the atmospheric profile is the same. You are simply converting between quantitities based on the P-T levels.

P2 L46: you may want to cite the Yamamoto-Wark paper as first historical record of CTP retrieval from oxygen absorption.

P2 L49-50: "Many approaches are designed to retrieve clouds' effective top pressures without considering their in-cloud photon penetration, and therefore derive effective top pressures higher than CTP."

I have two remarks for this statement.

1) there are other approaches taking into consideration in-cloud photon penetration. They must be correctly cited. Notably, analytical radiative trasnfer has been implemented by Kokhanovsky and Rozanov, JQSRT 2004 (forward problem) and Rozanov and Kokhanovsky, JGR 2004 (inverse problem) and globally deployed and validated by Lelli et al, AMT, 2012 and Lelli et al. ACP 2014. For the LUT method, the reference is Loyola et al. AMT 2018. So please, cite this literature.

2) The authors assume that the reader already knows the scientific reasoning behind the CTP overestimation / CTH underestimation. Which might not be true. So, please, explain here why the neglection of photon penetration and multiple scattering within the cloud gives rise to this effect.

P2 L 67-68: "the differences between in-band and reference band are negligible". This statement cannot be generalized. So, please add "at nominal EPIC response functions" or similar.

P3 L 86-87: "the ratios of absorption/reference are less impacted by the instrument calibration and other measurement error." I might agree with this statement if the authors can provide at least a reference to some EPIC assessment reports or papers where absolute (nor relative neither ratioed) calibration and degradation of the NIR channels are provided. I tend to believe it is the case but I would like to have this information at hand for sake of consistency.

Still Section -2- does not mention any surface influence. We know that the continuum at 779 nm is impacted by the red edge, whereas the b-band is not. So, I find myself left with the doubt: are the authors aware of this?

P3 Figure 1: Can the authors provide here in the caption or in the text the details of the simulation for these oxygen spectra? Mainly observational geometry, aerosol total load, ozone concentration and surface reflectivity/albedo?

P4 L106-107: "Cloud pressure thickness can be estimated with cloud optical thickness using statistical rules." Which are? Can the authours explain what statistical rules are they referring to and the physical principles behind this statement? References are also welcome along the way (this remark has to be read jointly with the remarks for Section 4.4 below).

P4 L 108-110: "It is worth noting that certain variables will have a non-linear effect on EPIC observations, however, these variations occur smoothly." Well, never poke a bear: could you please explain what are the variables smoothly having a non-linear effect on EPCI observations? First, what observations? Second, are these variables of radiometric or geometric origin? Are they clouds themselves? What kind of non-linear relationship are the authors thinking at? And if it a smooth one, this means it has been already well charachterized. Would you provide some figures or references as well?

P4 L114-116: "In physics, the retrieval accuracy is impacted by two main uncertainty sources: (1) the limited ability of EPIC in identifying cloud thermodynamic phase, which will affect the accuracy of cloud optical thickness retrieval, and 2) the uncertainty in

estimating Cloud pressure."

Yes correct. But this is disconnected from the sentence above about the interpolation error and the sentence here reads as a filler. So, I suggest to either expand this paragraph and describe throughly how the total error in CTP splits into random and systematic components, model and retrieval errors, and what originates them or, please, remove this sentence. Also because Section 3.1 is just about the LUT method. Ah, by the way, it would be very insightful to substantiate with numbers or references the LUT interpolation error component. Your choice.

P5 L145-146 and ff: "However, their attenuations from Rayleigh scattering and aerosol extinction are close to each other. Thus ... " I am personally not satisfied by these reoccurring statements in the manuscript. Too general, subjective and overly simplyfing. As such, the inference that photon path length can be derived by ratioing continuum and in-band channels does not follow from that. If you invert the logic, would the converse hold? Saying that molecular and aerosol extinction are not "close to each other" would still CTP retrieval be feasible? I would say it does. So, the issue here is that the authors simply avoid aerosol description for the sake of simplicity, but it is not what one would expect from an algorithm.

P5 L149-151: Please, refrain from wording like "and etc." and try to be rigorous. Assumptions are fine, as long as they are clearly presented and justified by a scale analysis or a scientific reasoning. So, please enumerate all assumptions you make and justify each of them.

P5 and ff: could you please use the standard \tau symbol for optical depth throughout the paper? \t can be misinterpreted as transmission.

P7 L215: missing to introduce the \k_i in the text. Please, correct.

P7 L222 and ff: How does Eq.14 relate to the conversion between CTP and CTH? Please, expand and/or reword this paragraph clearly exposing the practical usage of

this relationship w.r.t. cloud parameters to be retrieved. Also, what are the $\M_i$ (i=1...6) model atmospheres? Are you subsetting a yearly cycle in six different model atmospheres? Are you slicing after zonal bands?

P8 Equation 16: please be rigorous and consistent through the paper. Here you use $\t$ as temperature, while $\t$ was optical depth in the previous sections. So, temperature is $\T$, optical depth is $\tau$. Also, capital $\H$ is not present in the equation.

For the time being let me assume that the y-axis displays the following quantity: 100*(LBL - DBL_K)/LBL.

Also, without information about aerosol in the simulations, these results indicates that molecular scattering introduces a systematic bias, as can be seen in the continuum outside absorption. For the in-band channels, however, the sign of the residuals reverses. This points to a different treatment of oxygen layered extinction. From the perspective of the CTP retrieval, what counts is the ratio of the channels. Given Fig.2 and the definition of the residuals introduced above, my guess is that you are overestimating molecular scattering and underestimating oxygen absorption.

This translates into a quenched ratio between continuum and in-band channel than it is in reality, so that you will introduce a retrieval bias, because you will assign less oxygen absorption to the EPIC measurements and your CTP_top will be lower (or CTH_top higher).

I admit that after convolution with the instrument response function you might be less prone to this, but then I would appreciate also such values in Table 1, together with the same values for the A-band wavelengths.

In summary:

- please expand Table 1 with results for a Thick Cloud (which optical depth?) - provide also the altitude/pressure of the simulated thin and thick cloud (ensure that you have a representative altitude for the specific cloud: low-level thick cloud and high-level thin

cloud) - Specify if the thermodynamic phase of the thin cloud is mixed or ice. Assuming the low-level thick cloud is warm, aka liquid. - Present results for all 4 EPIC channels (680, 688, 764, 779 nm) seperately *AFTER* convolution with the EPIC narrowband functions - It is not clear to me what is the last column about. Is the Difference (+0.08%, -0.02%) the average relative difference across the band or only at 688 nm? As such, these numbers are little informative.

P10 L329: You might be correct about the similar behaviour of the A-band compared to the b-band. However, the presence of the red edge beyond 690 nm would make your results different for Figure 3-d. The authors suggest to have already such results for the A-band as well, so could you please create a separate Figure with only the dependence on surface albedo with the A and b-band together? This is more informative to the reader in general, as there are several instruments not convering the b-band but solely the A-band.

P11 Section 4.4 "Case studies ... "

This section is missing some important information and is disappointing to read because it lacks a clear structure and explanation of the results is not satisfying. I have several remarks.

Beside some corrections listed in the "Minor Comment" section, I wonder why are the authors introducing Eq.(15) about COT while ending the introducing paragraph with considerations about CTP retrieval.

Nevertheless, first, it is not clear where the data for Figure 4 come from. Please add a source repository to enable the replication of your results. It is not clear what L1 data are you processing. So, please give information on the timestamp and the data versioning, reprocessing and so on and guide the reader to the actual source, as not everyone ought to be fluent in EPIC data acquistion and handling.

Second, are the retrievals of Figure 4 for the full EPIC disc? The scatterplots show

clustering that must be analysed and understood. So, I invite the authors to subset L1 radiances after underyling surface reflectance and cloud optical thickness, or latitude or cloud system/regime so that you will be able to geophysically explain the scatter-plots. Also, in absence of bias histograms, they must be at least redrawn as heat or occurrence maps with a color coding for the third axis.

Third, Figure 4: you are comparing an "effective" CTP retrieval (the NASA ASDC L2 record) that does not include photon penetration with your "baseline" CTP method, which does not include photon penetration either. And you still have mean biases for low-level clouds of 100 mb and 150 mb for the A-band and b-band respectively. The apparent "banana" shape, bending toward the ground, might also indicate that you are using different P-T atmospheric profiles, which then impact gaseous extinction. Have you ensured that you are using the same atmosphere of the standard L2?

Fourth, I hope that the authors would agree with me that the results of Section 4 are still simply a verification of their algorithm and cannot be considered a real validation of their method. Figure 4 compares tow similar methods (as stated by the authors at P11 L335-336) while Figure 5 is simply an internal check of the methods presented in the paper. These results are already known in the literature bulk of A-band algo-rithms (e.g. by comparison of SACURA, FRESCO, ROCINN, See the TROPOMI S5P Science Verification Report). So, to gain insight in the validity and limitation of your algorithm and to let the reader decide whether your approch is best suited for a cloud type or another (for instance low-level warm or high-level thin cirrus clouds) indepen-dent validation is needed and must be carried out against a different CTP derived from coincident retrievals and alternative methods, being this ground-based or space-borne, your choice. But validation is needed.

Fifth, can the authors provide the reasoning behind the choice of their "statistical ap-proach" to estimate cloud geometrical/pressure thickness? Why are you calling it a statistical approach, I would rather call it assumption. Surely this assumption is based on evidence, likey drawn by references or assessment studies. So, please make the

derivation of your assumption about this approach explicit. Moreover, no details on the physics behind are given. Where are all the terms of the expression (i.e. the multiplicative factor 2.5, the additive +26) coming from? Expected limitations and range of applicability of this assumption? Any relationship with/dependence on cloud liquid water content and/or cloud type? One pertinent reference on my own I can come up with is Carbajal Henken et al. AMT, 2015 where CTP is related to pressure thickness and optical depth. But the same result has been obtained also by Rozanov and Kokhanovsky, JGR 2004 and Lelli et al, AMT, 2012 and ACP 2016 (see Appendix). It will be interesting to augment this bulk of literature with the references provided by the authors.

Finally, Figure 5. Fig. 5-a and 5-b extend the results of Fig.4-c and Fig.4-d, correct? You are using the same scenes of the NASA ACDC L2 record and you compare your baseline-CTP with the retrieved-CTP? Could you please elaborate why is the B-band closer to the A-band retrieval when photon penetration in the cloud is allowed? The sentence at P13 L378 ("This indicates, as expected, more photon penetration correction for B-band than A-band") reads a gap filler and sounds like tha authors want to get away with this without further investigation. There is a reason why the B-band is not customarily used for calibration of surface pressure. Some of the co-authors are surely aware of this effect.

Section 4.5 "Retrieval of global observation"

It is not clear if the same filtering (cloud cover = 1, cloud optical thickness $\geq$ 3, surface albedo < 0.25) is applied for the generation of the RGB snapshot of Fig.6-a. Also in view of Fig.6-d, COT: based on the visual inspection of the patterns, the cloud systems are quite different between the two maps, which are in turn also different from the CTP maps. The patterns are, in my opinion, quite different: the Nothern Pacific system is captured neither in the COT (Fig.6-d) nor in the CTP (Figs.6-b,c,e,f), being the B-band overall shallower/fainter than the A-band. This could point to the choice of grounding all filtered NANs (not-a-number) to 1013 mb, making them valid retrievals in the color

scale, albeit representing a fake surface pressure. I would then make this point grey or white, in all Figs.6 b to f and leaving Fig.6-a untouched.

It is not clear to my why the authors are using the L2 COT from NASA ASDC and not their own as specified by Eq.(15). If the calculation of COT in this paper differs (or it is the same) from the one in Yang et al.(2013) this must be stated at the beginning of Section 4.4. Otherwise the reader cannot judge in any way the soundness of the sentence in P13-14 L399-402 about the error propagation of COT into CTP.

To conclude, this section lacks some explanation about the patterns we see in the disc. I understand that the Pacific is a favourable geophysical scene to analyze, due to the lack of difficult reflective ground. However, the authors are capturing a wealth of cloud systems: deep convective clouds within the tropical belt, subsidence clouds in the trade wind belts, near-polar clouds at high latitudes, low-level warm cloud decks, even some cirrus clouds may slip through a COT filter of 3 (perhaps). Each of this cloud type can be categorized after its average cloud optical thickness. Please, introduce COT in your error analysis.

And also create difference maps centred on 0 mb with a divergent color palette for Fig.6c-Fig.6f and Fig.6c-NASA_L2_ASDC.

P14 L410 Conclusions.

- There is no Yuekui et al. 2012 in the bibliography. Please check.

- Here the authors need not just to summarize what they have done but also discuss in a compact way the results and highlight limitations of their method and future developments.

*** Minor comments

P1 L15: was -> is

P9 Figure 2: Please, define in the caption how the difference in reflectance is defined.

[Figure]

P10 L299: "sensibility of every variant"? You mean "sensitivity to every variable"?

P10 Figure 3: in the caption please specify that "umu" is cosine of SZA.

P10 L309: "ratio of upward diffuse ... ", missing a word, perhaps radiance or radiation?

P10 L318: please refrain from subjective statements such as "This is easy to understand".

P10 L327: you mean "thick" cloud and not "heavy" cloud?

P11 L338: if the baseline-CTP mothod is adopted, then in-cloud penetration is not "ignorable" but "ignored" instead. "Ignorable" suggests the existence of an option to be chosen, such that the method still enables the calculation of in-cloud penetration but the authors choose otherwise. "Ignored" implies that the method offers no option other than those provided. So, "ignored" is more rigorous and exact.

Section 4.4, Figures 4 and 5: control axis labels. "Pressure" not "Pressue".

P11 L339: "light reached cloud top is assumed". missing "that"

P12 L371: what do you mean here with the word "interaction"?

** References

Yamamoto, G. and Wark, D. Q.: Discussion of letter by A. Hanel: determination of cloud altitude from a satellite, J. Geophys. Res., 66, 3596, 1961.

Loyola, D. G., Gimeno García, S., Lutz, R., Argyrouli, A., Romahn, F., Spurr, R. J. D., Pedergnana, M., Doicu, A., Molina García, V., and Schüssler, O.: The operational cloud retrieval algorithms from TROPOMI on board Sentinel-5 Precursor, Atmos. Meas. Tech., 11, 409–427, https://doi.org/10.5194/amt-11-409-2018, 2018.

Verification of cloud top height, optical thickness and aerosol layer height, in "Sentinel-5P TROPOMI Science Verification Report, S5P-IUP-L2-ScVR-RP, Issue 2.1", Sect. 13.4-14.4, https://earth.esa.int/documents/247904/2474724/Sentinel-5P-

TROPOMI-Science-Verification-Report, 2015

Rozanov, V. V. and Kokhanovsky, A. A.: Semianalytical cloud retrieval algorithm as applied to the cloud top altitude and the cloud geometrical thickness determination from top-of-atmosphere reflectance measurements in the oxygen A band, J. Geophys. Res., 109, 4070, doi:10.1029/2003JD004104, 2004.

Kokhanovsky, A. A. and Rozanov, V. V.: The physical parameterization of the top-of-atmosphere reflection function for a cloudy atmosphere–underlying surface system: the oxygen A-band case study, J. Quant. Spectrosc. Rad. Tran., 85, 35–55, doi:10.1016/S0022-4073(03)00193-6, 2004.

Lelli L, Kokhanovsky, A.A., Rozanov, V.V., Vountas M., J.P Burrows: Linear trends in cloud top height from passive observations in the oxygen A-band, Atmospheric Chemistry and Physics, 14, 5679-5692, doi:10.5194/acp-14-5679-2014, 2014

Lelli L, Kokhanovsky, A.A., Rozanov, V.V., Vountas M., Sayer, A.M., J.P Burrows: Seven years of global retrieval of cloud properties using space-borne data of GOME, Atmospheric Measurement Techniques, 5, 1551-1570, doi:10.5194/amt-5-1551-2012, 2012

Carbajal Henken, C. K., Doppler, L., Lindstrot, R., Preusker, R., and Fischer, J.: Exploiting the sensitivity of two satellite cloud height retrievals to cloud vertical distribution, Atmos. Meas. Tech., 8, 3419–3431, https://doi.org/10.5194/amt-8-3419-2015, 2015

---

## Referee Comment (RC2) · Anonymous Referee #3 · 19 Jun 2020

This paper introduces a method to retrieve cloud top heights from measurements in the wavelength range ~680nm to ~780nm in and next to the oxygen A and B absorption bands. Measurements are performed by the EPIC sensor which is operated on a satellite near the first Sun-Earth Lagrange point so that scattering angles are always 165° or larger.

I agree with each point raised by the first reviewer. While the science is probably sound as far as can be judged from the current manuscript, the manuscript requires major revisions and a further round of review before it might be published as a final paper.

Besides some language issues, the description should be improved, e.g. not all steps in section 3.2 can be followed. Section 4 could be split in two parts, since the first part is more about method description while the second part shows the results. Maybe

Sect. 2 + 3 + the first half of Sect. 4 could be merged into one section (called 'Theory and methods' or just 'Methods') with several subsections. A discussion of the results is missing. The conclusion section currently is more like a summary. A few minor remarks:

- Line 14: " analytic transfer model ": Do you mean your retrieval? In my view, even if it is a relatively simple retrieval and the term 'model' may not be completely wrong it should be called retrieval (or inversion or maybe 'inverse model' or 'retrieval using a analytic transfer model' or similar) because at least some readers will connect the term 'model' more with a forward model than with a retrieval.

- Line 22: "a one-hundred-fold time reduction": Which time is reduced? (Computation time I guess) Compared to what? (line-by-line calculations?)

- Line 36: The spatial resolution of the sensor could be mention here. Also the scattering angle range (>=165°) could be mentioned somewhere.

- Figure 1 caption: The model should be mentioned here. Currently it is mentioned only later in the text. Is the figure for 1013hPa? Is it only for O2 or for all atmospheric constituents?

- Line 122: 'we are trying to develop' could be replaced by 'we develop'.

- Line 134: 'outer space' could be replaced by 'TOA'.

- Line 144: 'airmass and aerosol that located above or below cloud': also inside a cloud Rayleigh scattering and extinction by aerosols can happen.

- Line 152: 'between solar and satellite sensors': You mean 'between Sun and satellite sensor'?

- Line 154: 'layerd' should be 'layered'.

- Line 284: 'and hard to tell directly' should be removed.

- Line 371: 'decrease' should be 'increase' if I understand correctly.
* * *

---

## Author Comment (AC1) · 31 Jul 2020

We thank the Reviewers for their very thorough and constructive comments, which have helped to improve the quality of this paper. Below are our responses to their comments. The response (e.g., blue) follows each comment.

**Comments from the editors and reviewers:**

This paper introduces an algorithm for the determination of the cloud top pressure inferred from measurements of oxygen absorption in the NIR by the EPIC sensor onboard the DSCOVR platform. The topic is important and appropriate for the journal.

The paper shows some sound science and the authors have structured their manuscript in the correct way. All major sections needed to present a retrieval algorithm are, in my opinion, addressed. However, major improvements are still needed, and I will be willing to evaluate a revised version of the paper. I bullet-list improvements and remarks "general comments" section and then I delve in the explanation of specifics later on.

\*\*\* General main comments

- Before any scientific content scrutiny, I suggest to thoroughly check punctuation, syntax and word C-(c)-apitalization and arrangement prior publication. Copernicus service should definitely help here, but also and foremost checks by the English native coauthors. Uneven sentences or awkward wording are present throughout the manuscript and are too many to be listed by a referee. This will help to showcase the logic of the method and the importance of the results.

- In the introduction state clearly and make explicit the difference with Yang et al. JQSRT, 2013. As both papers share the same goal, data source and co-authors, it is important to highlight the advancement achieved in this paper with respect to previous literature. Some scientific insights about the difference between the A and the B-band are given in Yang et al. but are put to little of any use in this work. One would expect some science advancement and not a mere application or repetition of a method. All my criticism and required improvements naturally follow from this remark.

Author reply: In the revised manuscript, we have added some sentences to describe the scientific insights about Yang et al. JQSRT, 2013, which are shown as follows:

"...By using EPIC reflectance ratio data at oxygen A-band and B-band absorption to reference channels, Yang et al. (2013) developed a method to retrieve CTH and cloud geometrical thickness simultaneously for fully cloudy scene over ocean surface. First their method calculates cloud centroid heights for both A- and B-band channels using the ratios between the reflectance of the absorbing and reference channels, then derives the CTH and the cloud geometrical thickness from the two dimensional look up tables that relate the sum and the difference between the retrieved centroid heights for A- and B-bands to the CTH and the cloud geometrical thickness. The difference in the  $O_2$  A- and B-band cloud centroid heights is resulted from the different penetration depths of the two bands. Compared to the cloud height variability, the penetration depth differences are much smaller and the retrieval accuracy from this method can be affected by the instrument noise (Davis et al. 2018a, b)."

Compared to Yang et al. (2013), this paper uses an analytical method to address the issue of incloud penetration. This approach is less prone to errors caused by instrument noise and the results are more robust. In this paper, the analytical method "adopted ideas of the semianalytical model (*Kokhanovsky and Rozanov, 2004; Rozanov and Kokhanovsky, 2004*), and developed a quadratic EPIC analytic radiative transfer equation to analyze the radiative transfer in oxygen A- and B-band channels."

- Therefore, the treatment of aerosols is overly simplified or neglected together with error analysis as function of cloud optical thickness or cloud cover, since we know that from the remote sensing perspective these two quantities are connected.

Author reply: In the revised paper, we have added some comments to describe the aerosol extinction issue. The detailed information is shown in the replies of later questions.

- The coefficients A, B, C (P6 L199) must be presented otherwise the reader is not equipped with the knowledge to replicate the results.

Author reply: We have presented coefficients A, B, C in Equation 13 in the revised paper.

- The presentation and analysis of the results is suboptimal. Without proper and customary validation with external independent data sets little knowledge can be won about the applicability ranges of the presented method in real geophysical scenarios, which is one of the stated goals of the paper, otherwise Section 5 would not be presented.

Author reply: In the revised paper, we have added a new subsection 3.2, i.e., validation of the retrieval method. We used the cloud layer top pressure information from CALIPSO measurements as a reference to validate our retrieval method. Through the case of validation, we obtain the following results: "... under single layer cloud situations, the CTPs derived from EPIC measurements are close to the CTP from CALISPO; under multi-layer cloud situations, the CTP derived from EPIC measurements are larger than the CTP from CALISPO.... For these single layer cloud cases, the mean values of CTP of CALIPSO, EPIC effective, EPIC baseline and EPIC retrieval are 846, 834, 866 and 850 mb, respectively. Compared to the CTP from CALIPSO measurements , the EPIC effective and baseline CTPs are 12 mb smaller or 20 mb larger, respectively; the EPIC retrieval with consideration of photon penetration is only 4 mb larger. This shows that our method for the CTP retrieval is valid and accurate under single layer cloud situations, which can be detected by CALIPSO but hard to derive by our retrieval method. It is because the EPIC retrieved CTP mainly shows the pressure of cloud layer that reflects the major part of incident sun light."

\*\*\* Specific comments to individual sections

- Abstract

P1 L28: why "obviously"? It is not a straightforward inference and it is not objective, but subjective instead. Please, remove it from the abstract.

Author reply: We have removed it as suggested.

P1 L29-30: could you provide quantitative figures for the comparisons? Something like "Out of N cases, we found an average bias between CTP b- and A-band of xxx hPa\_xxx hPa\_stdv".

Author reply: We have added this sentence into the abstract as suggested.

"...Out of around 10000 cases, in retrieved CTP between A- and B-bands we found an average bias of 93 mb with standard deviation of 81 mb."

P2 L44: "their atmospheric profiles"? You may want to check this, because the atmospheric profile is the same. You are simply converting between quantities based on the P-T levels.

Author reply: We have revised it to "the related atmospheric profile".

P2 L46: you may want to cite the Yamamoto-Wark paper as first historical record of CTP retrieval from oxygen absorption.

Author reply: We have cited the Yamamoto-Wark paper as suggested.

P2 L49-50: "Many approaches are designed to retrieve clouds' effective top pressures without considering their in-cloud photon penetration, and therefore derive effective top pressures higher than CTP."

I have two remarks for this statement.

1) there are other approaches taking into consideration in-cloud photon penetration. They must be correctly cited. Notably, analytical radiative transfer has been implemented by Kokhanovsky and Rozanov, JQSRT 2004 (forward problem) and Rozanov and Kokhanovsky, JGR 2004 (inverse problem) and globally deployed and validated by Lelli et al, AMT, 2012 and Lelli et al. ACP 2014. For the LUT method, the reference is Loyola et al. AMT 2018. So please, cite this literature.

Author reply: In the revised paper, we have cited these literatures and another paper Richardson and Stephens (2018):

"Although the theory of using oxygen absorption bands to retrieve CTP was proposed decades ago (Yamamoto and Wark, 1961), it is still very challenging to do the retrieval accurately due to the complicated in-cloud penetration effect (Yang et al., 2019, 2013; Davis et al., 2018a, 2018b; Richardson and Stephens, 2018; Loyola et al., 2018; Lelli et al., 2014, 2012; Schuessler et al., 2013; Rozanov and Kokhanovsky, 2004; Kokhanovsky and Rozanov, 2004; Kuze and Chance, 1994; O'brien and Mitchell, 1992; Fischer and Grassl, 1991; and etc.).... In the meantime, to improve the retrieval accuracy of CTP, various techniques have been applied to the retrieval methods with in-cloud photon penetration. For example, Kokhanovsky and Rozanov (2004) proposed a simple semi-analytical model for calculation of the top-of-atmosphere (TOA) reflectance of an underlying surface-atmosphere system, accounting for aerosol and cloud scattering. Based on the work of Kokhanovsky and Rozanov (2004), Rozanov and Kokhanovsky (2004) developed an asymptotic algorithm for the CTH and the geometrical thickness determination using measurements of the cloud reflection function. This retrieval method was

applied by Lelli et al. (2012, 2014) to derive CTH using measurements from GOME instrument on board the ESA ERS-2 space platform."

Richardson, M. and Stephens, G.L., 2018. Information content of OCO-2 oxygen A-band channels for retrieving marine liquid cloud properties. *Atmospheric Measurement Techniques*, *11*(3), pp.1515-1528.

2) The authors assume that the reader already knows the scientific reasoning behind the CTP overestimation / CTH underestimation. Which might not be true. So, please, explain here why the neglection of photon penetration and multiple scattering within the cloud gives rise to this effect.

Author reply: In the revised paper, we have added the following two sentences:

"...To estimate the CTP from satellite measurements, many approaches have been designed to retrieve clouds' effective top pressures without considering in-cloud photon penetration. These approaches did not consider light penetrating cloud, therefore the derived CTH is lower than the cloud top, and the effective top pressures is higher than CTP...."

P2 L 67-68: "the differences between in-band and reference band are negligible". This statement cannot be generalized. So, please add "at nominal EPIC response functions" or similar.

Author reply: We have revised it as suggested.

P3 L 86-87: "the ratios of absorption/reference are less impacted by the instrument calibration and other measurement error." I might agree with this statement if the authors can provide at least a reference to some EPIC assessment reports or papers where absolute (nor relative neither ratioed) calibration and degradation of the NIR channels are provided. I tend to believe it is the case but I would like to have this information at hand for sake of consistency.

Author reply: Currently, we do not find the exact statements from reference literature to present this comment. But we can draw the conclusion from the studies of Marshak et al. (2018).

Marshak, A., J. Herman, A. Szabo, K. Blank, S. Carn, A. Cede, I. Geogdzhaev, D. Huang, L.-K. Huang, Y. Knyazikhin, M. Kowalewski, N. Krotkov, A. Lyapustin, R. McPeters, K. Meyer, O. Torres and Y. Yang, 2018. Earth Observations from DSCOVR/EPIC Instrument. *Bulletin Amer. Meteor. Soc. (BAMS)*, 9, 1829-1850, https://doi.org/10.1175/BAMS-D-17-0223.1.

According to this paper, the calibration of EPIC measurements consists of two steps: (1) From Level-0 data, raw EPIC data (counts per second), to Level-1A "corrected count rates". This step includes 6 steps, such as dark offset correction, nonlinear correction, temperature correction, stray-light corrections, and etc. (2) Geolocation algorithms from Level 1A to Level 1B. To convert the count rates to reflectance data, calibration factors are needed. The reflectance calibration is implemented by using other satellite instruments like OMPS and MODIS. For example, EPIC 680 and 780 nm channels use MODIS to obtain calibration factor  $K_{\lambda}$ . For oxygen A-and B-band channels, lunar observations are used for calibration. "Lunar reflectance  $R_{\lambda}$  does

not increase much with a small wavelength change  $\Delta \lambda$ ; a 10-nm difference in  $\lambda$  leads to a difference in  $R_{\lambda}$  in the range of 0.0006–0.0013 or 0.8%–1.2% (e.g., Ohtake et al. 2010, 2013). ...."

"...Indeed, the ratio  $F(\lambda_1, \lambda_2)$  of the lunar reflectance values measured in counts per second at two neighboring channels  $\lambda_1$  and  $\lambda_2$  is very stable...." The calibration factors of 688 and 764 nm are calculated as follows (Geogdzhayev and Marshak, 2018):

Geogdzhayev, I. and A. Marshak, 2018. Calibration of the DSCOVR EPIC visible and NIR channels using MODIS Terra and Aqua data and EPIC lunar observations. *Atmos. Meas. Tech.* 11, 359 -368, https://doi.org/10.5194/amt-11-359-2018

$$K_{688} \approx \frac{K_{680}}{F(680,688)}; \ K_{764} \approx \frac{K_{780}}{F(780,764)}$$

From the above calibration processes, we can get the following information:

- (1) For EPIC oxygen A- and B-band channels, we need a series of calibration to obtain the reflectance data, which accuracy is impacted by many factors. Take 688 nm channel, as an example; its accuracy is impacted by the preprocessing calibration error, accuracy of  $K_{680}$  and F(680,688).
- (2) For the ratio of absorption/reference (e.g.,  $R_{688}/R_{680}$ ): because all EPIC channels share the same optical system and the same CCD sensors, some preprocessing calibration errors can be reduced when we calculate the ratio of two channels. For the  $R_{688}/R_{680}$ , the impact of accuracy of  $K_{680}$  is eliminated, because it is only determined by  $\frac{K_{680}}{K_{688}}$  or F(680,688).

Therefore, we can say: "the ratios of absorption to reference channels are less impacted by the instrument calibration and other measurement error." We have updated the manuscripts as follows:

"...Also, compared to any specific EPIC oxygen absorption bands (i.e.,  $R_{764}$  and  $R_{688}$ ), the ratios of absorption to reference channels (i.e.,  $R_{764}/R_{779}$  and  $R_{688}/R_{680}$ ) are less impacted by the instrument calibration and other measurement error. This can be explained by the following reasons: First, the EPIC measurements at oxygen A and B absorption and reference bands share same sensor and optical system, when calculating the ratios of them, some preprocessing calibration errors can be reduced. Second, to calculate  $R_{764}$  and  $R_{688}$ , the ratio of lunar reflectance at neighboring channels (i.e., F(764,779) and F(688,680)) and the calibration factors of oxygen A and B reference bands (i.e.,  $K_{779}$  and  $K_{680}$ ) are used (Geogdzhayev and Marshak , 2018; Marshak et al., 2018). Therefore, the accuracy of  $R_{764}$  and  $R_{688}$  is determined by the stability of F(764,779) and F(688,680) and the accuracy of  $K_{779}$  and  $K_{680}$  together. But the accuracy of absorption to reference ratios is only determined by the stability of (764,779) and F(688,680)."

Still Section -2- does not mention any surface influence. We know that the continuum at 779 nm is impacted by the red edge, whereas the b-band is not. So, I find myself left with the doubt: are the authors aware of this?

Author reply: Thank you for reminding the issue of red-edge. We have added a comment about it into the revised paper as follows: "...It is worth noting that for EPIC measurements at both

oxygen A- and B-bands, the surface influence cannot be ignored. For examples, in the snow or ice covered area the surface albedo is high; in the plants covered area, the surface albedo changes substantially between oxygen A-band and B-band due to the impact of spectral red-edge (*Seager et al., 2005*)."

P3 Figure 1: Can the authors provide here in the caption or in the text the details of the simulation for these oxygen spectra? Mainly observational geometry, aerosol total load, ozone concentration and surface reflectivity/albedo?

Author reply: In Figure 1, the absorption optical depth spectrum at the oxygen A and B bands is only related to the oxygen absorption coefficients and the atmospheric model. We have added some detailed information about simulation into the revised paper, which is shown as follows:

"...The high resolution absorption optical depth spectrum at oxygen A-band and B-band is calculated by Line-By-Line Radiative Transfer Model (LBLRTM, *Clough et al., 2005*) with HITRAN 2016 database (*Gordon et al., 2017*) for the U.S. standard atmosphere."

P4 L106-107: "Cloud pressure thickness can be estimated with cloud optical thickness using statistical rules." Which are? Can the authors explain what statistical rules are they referring to and the physical principles behind this statement? References are also welcome along the way (this remark has to be read jointly with the remarks for Section 4.4 below).

Author reply: In this study, the retrieval method cannot retrieve the cloud pressure thickness with CTP simultaneously, and it considers the cloud pressure thickness as an input parameter for CTP retrieval. We will improve our method to address this issue in the future.

Currently, we use cloud optical thickness to estimate cloud pressure thickness by using NASA atmospheric reanalysis data. In the revised manuscript, we have added detailed comments to state how we use cloud optical thickness to estimate the cloud pressure thickness:

"...The cloud pressure thickness or the cloud vertical distribution has substantial impact on the accuracy of the CTP retrievals (*Carbajal Henken et al., 2015; Fischer and Grassl, 1991; Rozanov and Kokhanovsky, 2004; Preusker and Lindstrot, 2009*). In this study, the cloud pressure thickness is used as an input parameter to retrieve the CTP. However, no related accurate cloud pressure thickness is provided by other satellite sensors now. To constrain the error from the estimation of cloud pressure thickness, we relate it to the cloud optical thickness. It is reasonable because clouds with higher optical thickness normally have higher values of pressure thickness. To explore the correlation between cloud pressure thickness and cloud optical thickness, we use the related cloud data from Modern-Era Retrospective analysis for Research and Applications Version 2 (MERRA-2, Gelaro et al., 2017), which is a NASA atmospheric reanalysis for the satellite era using the Goddard Earth Observing System Model Version 5 (GEOS-5) with Atmospheric Data Assimilation System (ADAS).

Based on statistical analysis of one year's single-layer liquid water clouds over an oceanic region (S23.20, W170.86, S2.11, W144.14) in 2017, we can get an equation for cloud pressure thickness approximation, i.e., cloud pressure thickness (mb) = 2.5\* COD + 23. The derived correlation coefficients are dependent on the case region and time selections. Due to the

complexity of cloud vertical distribution, whatever the accuracy of the correlation coefficients is, the estimation will certainly bring in error."

The scatter plot of cloud pressure thickness and cloud optical thickness is shown in Figure R1. (After re-checking the cloud data, we updated the equation as: cloud pressure thickness (mb) = 2.5\* COD + 23.)

Figure R1. The scatter plot of cloud pressure thickness and cloud optical depth and the related linear fitting line.

P4 L 108-110: "It is worth noting that certain variables will have a non-linear effect on EPIC observations, however, these variations occur smoothly." Well, never poke a bear: could you please explain what are the variables smoothly having a non-linear effect on EPCI observations? First, what observations? Second, are these variables of radiometric or geometric origin? Are they clouds themselves? What kind of non-linear relationship are the authors thinking at? And if it a smooth one, this means it has been already well characterized. Would you provide some figures or references as well?

Author reply: Maybe this sentence is ambiguous and make readers confuse. The original meaning of it is shown as follows: the "ratio of simulated reflectance measurements for EPIC absorption/reference" is a function of multiple variables, i.e., surface albedo, cloud optical depth, solar zenith angle, cloud top pressure and cloud pressure thickness. The effects of these variables on that ratio may be not linear, such as COD, as shown in Figure 3 in the manuscript. If the resolution of the LUT is high, we still can use linear interpolation method to retrieve the unknown variable with high accuracy. Take a simple example, for an exponential function  $y=\exp(x)$ , y is not a linear function of x, but if we have a series of pair values  $(x_i, y_i)$  in the range of x = [1,4] with high resolution (e.g., 0.02), we still can calculate exp (3.535) with pretty high accuracy by using linear interpolation method to exp(3.52) and exp (3.54). We have revised this sentence in the manuscript:

"...It is worth noting that the reflectance ratio of absorption/reference can be seen as a function of surface albedo, solar zenith and viewing angles, COD, CTP and cloud pressure thickness. Some atmospheric variables have a non-linear effect on the reflectance ratio. For example, the reflectance ratio is more sensitive to the variation of COD when COD is small. Overall, the reflectance ratio varies monotonically and smoothly with these variables (shown in Figure 3). With a relatively high-resolution simulated table, we can use a localized linear interpolation method to estimate the proper values..."

P4 L114-116: "In physics, the retrieval accuracy is impacted by two main uncertainty sources: (1) the limited ability of EPIC in identifying cloud thermodynamic phase, which will affect the accuracy of cloud optical thickness retrieval, and 2) the uncertainty in estimating Cloud pressure."

Yes correct. But this is disconnected from the sentence above about the interpolation error and the sentence here reads as a filler. So, I suggest to either expand this paragraph and describe thoroughly how the total error in CTP splits into random and systematic components, model and retrieval errors, and what originates them or, please, remove this sentence. Also because Section 3.1 is just about the LUT method. Ah, by the way, it would be very insightful to substantiate with numbers or references the LUT interpolation error component. Your choice.

Author reply: We have removed this sentence as suggested. We also added more comments about the LUT based approach with some references. Parts of the revised paragraph are shown as follows:

"One commonly used method of retrieval for satellite observation is through the building and usage of LUTs (Loyola et al., 2018, Gastellu-Etchegorry and Esteve, 2003). LUT based approach can be fast because the most computationally expensive part of the inversion procedure is completed before the retrieval itself. For DSCOVR EPIC observations, we can build LUTs by simulating EPIC measurements under various atmospheric conditions, such as different surface albedo, solar zenith and viewing angles, COD, CTP, and cloud pressure thickness. Comparing the related simulated reflectance at the oxygen absorption and reference bands, we can obtain two LUTs for reflectance ratios of absorption/reference at EPIC oxygen A-band and B-band respectively, which can be used for the CTP retrieval. The detailed information of simulated reflectance ratio of absorption/reference is stated in Section 2.3.3. ..."

"....The retrieval error of this method is determined by the resolution of the LUT, i.e., the higher the resolution, the higher retrieval accuracy. However, for multiple dimensional LUTs, the increase of resolution will increase the table size exponentially, which will increase computational cost substantially for the table building and inverse searching. Another possible method to increase the retrieval accuracy is using different interpolation methods. For example, if the value of LUT varies non-linearly with a variable, using high order interpolation method maybe better than using linear interpolation method (Dannenberg, 1998)."

P5 L145-146 and ff: "However, their attenuations from Rayleigh scattering and aerosol extinction are close to each other. Thus ... " I am personally not satisfied by these reoccurring statements in the manuscript. Too general, subjective and overly simplyfing. As such, the inference that photon path length can be derived by ratioing continuum and in-band channels

does not follow from that. If you invert the logic, would the converse hold? Saying that molecular and aerosol extinction are not "close to each other" would still CTP retrieval be feasible? I would say it does. So, the issue here is that the authors simply avoid aerosol description for the sake of simplicity, but it is not what one would expect from an algorithm.

Author reply: We have revised this sentence as follows:

"...Oxygen A-band and its reference band are also attenuated by airmass and aerosol through Rayleigh scattering and aerosol extinction. In the standard atmospheric model, the optical depth of Rayleigh scattering ( $\tau_{Ray}$ ) at oxygen A-band (B-band) and its reference band is 0.026 (0.040) and 0.024 (0.042), respectively (Bodhaine et al., 1999). The absolute difference of Rayleigh scattering optical depth ( $\Delta \tau_{Ray} = \tau_{Ray}^{In-band} - \tau_{Ray}^{Ref}$ ) between them is within 0.002. Compared to Rayleigh scattering, the difference of background aerosol optical depth ( $\Delta \tau_{Aer}$ ) between absorbing and reference bands is smaller, within 0.0005. Therefore, the attenuations from Rayleigh scattering and aerosol extinction at EPIC oxygen absorption and its reference band are close to each other. Thus, when we use the ratio of EPIC measured reflectance at oxygen A-band and its reference band to derive the photon path length distribution and retrieve cloud information such as CTP, the impact of Rayleigh scattering and aerosol extinction can be simplified in the analytic transfer inverse model."

We also revised Eq. (11) to show the impact of background extinction.

"Combining Eqs. (2), (9) and (10), we can get the total EPIC analytic transfer equation as follows

$$-log\left(\frac{R_A}{R_f}\right) = f\left(\Delta\tau_{O2}^{Above-Cld}, \mu_0, \mu, \varphi\right) + f\left(\tau_{O2}^{Top}, \Delta\tau_{O2}^{Cld}, \mu_0, \mu, \varphi\right) + f\left(\Delta\tau_{O2}^{Below-Cld}, \mu_0, \mu, \varphi\right) + \Delta\tau_{BG}\left(\frac{1}{\mu} + \frac{1}{\mu_0}\right)$$
(11)

In Eq. (11),  $\Delta \tau_{BG}$  represents the sum of optical depth difference of background extinction (i.e., Rayleigh scattering  $\Delta \tau_{Ray}$ , aerosol extinction  $\Delta \tau_{Aer}$ , and O3  $\Delta \tau_{O3}$ ) between oxygen in-band and reference band, as shown in Eq. (12).

$$\Delta \tau_{BG} = \Delta \tau_{Ray} + \Delta \tau_{Aer} + \Delta \tau_{O3} \tag{12}$$

As stated in the previous subsection, in the standard atmospheric model with background aerosol loading, ( $\Delta \tau_{Ray}$ ,  $\Delta \tau_{Aer}$ ,  $\Delta \tau_{O3}$ ) is approximately (0.002, 0.0005, -0.0005) and (-0.002, -0.0005, -0.0002) respectively at oxygen A and B bands, thus  $\Delta \tau_{BG}$  is approximately 0.002 and -0.0045 respectively at these two bands."

P5 L149-151: Please, refrain from wording like "and etc." and try to be rigorous. Assumptions are fine, as long as they are clearly presented and justified by a scale analysis or a scientific reasoning. So, please enumerate all assumptions you make and justify each of them.

Author reply: We have revised this sentence: "To simplify the analytic transfer inverse model for EPIC observations, we made a series of assumptions, e.g., isotropic component, a planeparallel homogenous cloud assumption with quasi-Lambertian reflecting surfaces. These assumptions have been widely used in radiative transfer calculation for cloud studies." P5 and ff: could you please use the standard \tau symbol for optical depth throughout the paper? \t can be misinterpreted as transmission.

Author reply: We have revised it, as suggested, by using  $\tau$  to replace t.

P7 L215: missing to introduce the  $k_i$  in the text. Please, correct.

Author reply: We have defined  $k_i$  in the revised manuscript: ... $k_i$  is the line shapes of oxygen A- and B-bands.

P7 L222 and ff: How does Eq.14 relate to the conversion between CTP and CTH?

Author reply: The Eq.16 (i.e., Eq.14 in the original manuscript) is used to calculate oxygen absorption coefficients for any given atmospheric profiles. It is not directly related to the conversion between CTP and CTH. In this paper, we mainly focus on the retrieval of CTP and all discussions are mainly focused on the CTP too. We have revised the related paragraph as follows:

"In the simulation of EPIC measurements, the atmospheric layer at a given layer-average pressure can have drastically different temperature depending on the atmospheric profile in use. To ensure the accuracy of simulation, we need to use the LBLRTM package to calculate oxygen absorption coefficients for each pressure/temperature profile, which is a time-consuming process. Our goal has been to find a simple and fast method to calculate oxygen absorption coefficients for different atmospheric profiles. Based on the study of Chou and Kouvaris (1986), Min et al. (2014) proposed a fast method to calculate oxygen absorption optical depth for any given atmosphere by using a polynomial fitting function, as shown in Eq. (16).

$$\ln (A_{vLM}) = [a_0(v, P) + a_1(v, P) \times (T_{LM} - T_{mL}) + a_2(v, P) \times (T_{LM} - T_{mL})^2] \times \rho_{O_2}$$
(16)

Where  $A_{vLM}$  is optical depths for layer L, spectral point v, and atmosphere model M;  $\rho_{0_2}$  is molecular column density  $(\frac{molecules}{cm^2} \times 10^{-23})$ ; TLM is the average temperature for layer L for a given atmosphere; and TmL is average temperature over all six typical geographic-seasonal model atmospheres (M1 to M6, i.e., tropical model, mid-latitude summer model, mid-latitude winter model, subarctic summer model, subarctic winter model, and the U.S. Standard (1976) model) for layer L. To derive the coefficients a0, a1, and a2, we first calculated oxygen optical depth coefficients for all typical atmospheres (M1 to M6) by using LBLRTM package, and then selected three of them (e.g., M1, M5, and M6) to calculate the polynomial fitting coefficients. This method has been successfully used by Min et al. (2014) to simulate the high resolution oxygen A-band measurements."

Please, expand and/or reword this paragraph clearly exposing the practical usage of this relationship w.r.t. cloud parameters to be retrieved. Also, what are the nM\_i (i=1...6) model atmospheres? Are you subsetting a yearly cycle in six different model atmospheres? Are you slicing after zonal bands?

Author reply: We did not subset a yearly cycle in six different model atmospheres or slice after zonal bands. We have revised this paragraph as shown in the answer to the last question.

"...Where  $A_{vLM}$  is optical depths for layer L, spectral point v, and atmosphere model M;  $\rho_{O_2}$  is molecular column density  $(\frac{molecules}{cm^2} \times 10^{-23})$ ;  $T_{LM}$  is the average temperature for layer L for a given atmosphere; and  $T_{mL}$  is average temperature over all six typical geographic-seasonal model atmospheres (M1 to M6, i.e., tropical model, mid-latitude summer model, mid-latitude winter model, subarctic summer model, subarctic winter model, and U.S. Standard (1976) model) for layer L. To derive the coefficients  $a_0$ ,  $a_1$ , and  $a_2$ , we first calculated oxygen optical depth coefficients for all typical atmospheres (M1 to M6) by using LBLRTM package, and then selected three of them (e.g., M1, M5, and M6) to calculate the polynomial fitting coefficients. This method has been successfully used by Min et al. (2014) to simulate the high resolution oxygen A-band measurements."

P8 Equation 16: please be rigorous and consistent through the paper. Here you use  $\$ t as temperature, while  $\$ t was optical depth in the previous sections. So, temperature is  $\$ T, optical depth is  $\$ tau. Also, capital  $\$ H is not present in the equation. For the time being let me assume that the y-axis displays the following quantity:

100\*(LBL - DBL\_K)/LBL.

Author reply: We have changed "t" to "T" to represent temperature. The capital H has been changed to h. The y-axis displays the relative difference:  $100*(DBL_K-LBL)/LBL$ .

Also, without information about aerosol in the simulations, these results indicate that molecular scattering introduces a systematic bias, as can be seen in the continuum outside absorption. For the in-band channels, however, the sign of the residuals reverses.

This points to a different treatment of oxygen layered extinction. From the perspective of the CTP retrieval, what counts is the ratio of the channels. Given Fig.2 and the definition of the residuals introduced above, my guess is that you are overestimating molecular scattering and underestimating oxygen absorption.

This translates into a quenched ratio between continuum and in-band channel than it is in reality, so that you will introduce a retrieval bias, because you will assign less oxygen absorption to the EPIC measurements and your CTP\_top will be lower (or CTH\_top higher).

I admit that after convolution with the instrument response function you might be less prone to this, but then I would appreciate also such values in Table 1, together with the same values for the A-band wavelengths.

Author reply: In the updated manuscript, Section 2.3.2 and Figure 2 describes the application of double-k approach based on the fast radiative transfer model. The detailed information about this fast radiative transfer model was shown in the Duan et al. JGR, 2005: "Duan, M., Min, Q. and Li, J.: A fast radiative transfer model for simulating high-resolution absorption bands. Journal of Geophysical Research: Atmospheres, 110(D15), 2005."

The bias in Figures 2 and R2 is from the accuracy of double-k approach itself. In both LBL benchmark simulation and fast radiative transfer model simulation, we already considered the

Rayleigh scattering and aerosol loading. In this study, background aerosol with AOD = 0.08 is used in the radiative transfer calculation.

Figure R2. Differences between simulated spectra by the benchmark and fast radiative transfer models as a function of absorption optical depths for a clear day case.

From our point of view, the error of CTP retrieval from radiative transfer calculation should be negligible. When absorption optical depth is small (out of band area), the relative difference is only around 0.1%. Although the relative accuracy of high resolution spectra at oxygen absorption peak positions between fast radiative transfer calculation and LBL calculation is up to 1%, but its effect on the radiation is very small because of the high OD at that wavelength position. After convolution with the instrument response function, the accuracy of the fast radiative transfer model is high, as shown in Table 1. The other thing is that we only used the double-k approach to calculate oxygen A- and B-band absorption channels (764 and 688 nm). For reference bands, we did radiative transfer calculation directly by using narrowband profiles of oxygen absorption optical depth at 679.64 and 779.24 nm.

The retrieval errors of CTP are mainly from other sources, we will discuss them in the replies for the later questions.

**In summary:**

- please expand Table 1 with results for a Thick Cloud (which optical depth?) – provide also the altitude/pressure of the simulated thin and thick cloud (ensure that you have a representative altitude for the specific cloud: low-level thick cloud and high-level thin cloud) - Specify if the thermodynamic phase of the thin cloud is mixed or ice. Assuming the low-level thick cloud is warm, aka liquid. - Present results for all 4 EPIC channels (680, 688, 764, 779 nm) separately \*AFTER\* convolution with the EPIC narrowband functions - It is not clear to me what is the last column about. Is the Difference (+0.08%, -0.02%) the average relative difference across the band or only at 688 nm? As such, these numbers are little informative.

Author reply: For EPIC oxygen A-band and B-band reference channels (779 and 680 nm), because their optical depth spectra are smooth and contain no absorption lines, we do not calculate high resolution spectra for them. We do calculations directly by using the narrowband oxygen absorption optical depth profiles at 679.64 and 779.24nm. In the revised manuscript, we expanded Table 1 for both oxygen A and B absorption bands, including results for a thick cloud. The last column shows the cases' relative error between double-k approach and LBL calculation:  $(DBL_K - LBL)/LBL*100\%$ . In this study, all the radiative transfer calculations are based on the assumption of homogenous liquid water cloud.

The expanded Table 1 and updated manuscript is shown as follows:

"...Therefore, for the simulated narrowband measurements at EPIC oxygen B-band, the relative difference between LBL and double-k approach is much smaller than that of the high resolution spectrum, which is less than 0.1% for clear day. Compared to clear sky situation, the relative difference for cloud situations can be bigger. As shown in Table 1, the relative difference is - 0.06% and -0.32% for typical high level optical thin cloud and low-level thick cloud situations, respectively. The comparison of simulated narrowband measurement at EPIC oxygen A-band channel (764 nm) is also shown in Table 1, the relative differences between LBL and double-k approach are -0.06%, 0.21% and 0.23% for clear day, high level thin cloud and low level thick cloud situations approach are spectively. In general, the accuracy of double-k approach for both oxygen A and B absorption bands is high."

| Case (SZA=35, surface albedo
=0.02)          |        | Line by Line | Double k | Relative
Difference |  |
|-------------------------------------------------|--------|--------------|----------|------------------------|--|
| Clear Day                                       | 688 nm | 0.026963     | 0.026985 | +0.08%                 |  |
|                                                 | 764 nm | 0.013979     | 0.013970 | -0.06%                 |  |
| Thin cloud
(COD=2, 8.3-
8.5 km, liquid)   | 688 nm | 0.098444     | 0.098131 | -0.32%                 |  |
|                                                 | 764 nm | 0.071359     | 0.071507 | +0.21%                 |  |
| Thick cloud
(COD=16, 1.5-
2.9 km, liquid) | 688 nm | 0.396354     | 0.396117 | -0.06%                 |  |
|                                                 | 764 nm | 0.233937     | 0.234485 | +0.23%                 |  |

| Table 1 | . Comp | oarison o | of simulated | l narrowband | measurement at | EPIC | A- and | B-Band | channels |
|---------|--------|-----------|--------------|--------------|----------------|------|--------|---------------|----------|
|---------|--------|-----------|--------------|--------------|----------------|------|--------|---------------|----------|

P10 L329: You might be correct about the similar behavior of the A-band compared to the bband. However, the presence of the red edge beyond 690 nm would make your results different for Figure 3-d. The authors suggest to have already such results for then A-band as well, so could you please create a separate Figure with only the dependence on surface albedo with the A and b-band together? This is more informative to the reader in general, as there are several instruments not convering the b-band but solely the A-band.

Author reply: In the simulation, we set a series of surface albedo for both oxygen A-band and B-band. However, when we calculate the ratio of absorption/reference, we assume that the oxygen

absorption band and reference band have the same surface albedo. If there is substantial difference of surface albedo between oxygen A-band and B-band due to the red edge, the retrieved CTP based on measurements of oxygen A-band and B-band may have a big difference if the impact of the red edge is not accounted.

We have added a separate Figure (Figure 4) with only the dependence on surface albedo with the A and B-band together. The figure and the related paragraph are shown as follows:

---

## Author Comment (AC2) · 31 Jul 2020

We thank the Reviewers for their very thorough and constructive comments, which have helped to improve the quality of this paper. Below are our responses to their comments. The response (e.g., blue) follows each comment.

**Comments from the editors and reviewers:**

This paper introduces a method to retrieve cloud top heights from measurements in the wavelength range ~680nm to ~780nm in and next to the oxygen A and B absorption bands. Measurements are performed by the EPIC sensor which is operated on a satellite near the first Sun-Earth Lagrange point so that scattering angles are always 165º or larger.

I agree with each point raised by the first reviewer. While the science is probably sound as far as can be judged from the current manuscript, the manuscript requires major revisions and a further round of review before it might be published as a final paper.

Besides some language issues, the description should be improved, e.g. not all steps in section 3.2 can be followed. Section 4 could be split in two parts, since the first part is more about method description while the second part shows the results. Maybe

Sect. 2 + 3 + the first half of Sect. 4 could be merged into one section (called 'Theory and methods' or just 'Methods') with several subsections. A discussion of the results is missing. The conclusion section currently is more like a summary. A few minor remarks:

Author reply:  Thank you very much for your comments, we have revised the structure of the paper as suggested : The original sections 2, 3 and half of section 4 have been merged into one section "Theory and methods", the other half of section 4 is categorized into another section "Application and validation of the CTP retrieval method".

- Line 14: " analytic transfer model ": Do you mean your retrieval? In my view, even if it is a relatively simple retrieval and the term 'model' may not be completely wrong it should be called retrieval (or inversion or maybe 'inverse model' or 'retrieval using a analytic transfer model' or similar) because at least some readers will connect the term 'model' more with a forward model than with a retrieval.

Author reply:  We have revised it to "An analytic transfer inverse model" as suggested. We also replaced the "analytic transfer model" by "analytic transfer inverse model"  in all other places in the paper.

- Line 22: "a one-hundred-fold time reduction": Which time is reduced? (Computation time I guess) Compared to what? (line-by-line calculations?)

Author reply:  We have revised this sentence as follows: "…To simulate the EPIC measurements, a program package using the double-$k$ approach was developed. Compared to line-by-line calculation, this approach can calculate high-accuracy results with a one-hundred-fold computation time reduction…."

- Line 36: The spatial resolution of the sensor could be mention here. Also the scattering angle range (>=165_) could be mentioned somewhere.

Author reply: We have revised as suggested: "…One of the Earthward instruments is the Earth Polychromatic Imaging Camera (EPIC) sensor, which can take images of the Earth with spatial resolution of 10 km at nadir. The EPIC continuously monitors the entire sunlit Earth for backscatter, with a nearly constant scattering angle between 168.5° and 175.5°, from sunrise to sunset with 10 narrowband filters: 317, 325, 340, 388, 443, 552, 680, 688, 764 and 779 nm (Marshak et al., 2018)…"

- Figure 1 caption: The model should be mentioned here. Currently it is mentioned only later in the text. Is the figure for 1013hPa? Is it only for O2 or for all atmospheric constituents?

Author reply: We have revised the Figure 1 caption as suggested:

"**Figure 1:** High resolution calculated absorption optical depth spectrum at oxygen A-band (a) and B-band (b) with DSCOVR EPIC oxygen A and B bands in-band and reference filters. Here the absorption optical depth spectrum is calculated by LBLRTM model with HITRAN 2016 database for the U.S. standard atmosphere."

This figure is for U.S. standard atmosphere, which surface pressure is 1013 hPa. It is for all atmospheric constituents.

- Line 122: 'we are trying to develop' could be replaced by 'we develop'.

Author reply: We have revised as suggested.

- Line 134: 'outer space' could be replaced by 'TOA'.

Author reply: We have revised as suggested.

- Line 144: 'airmass and aerosol that located above or below cloud': also inside a cloud Rayleigh scattering and extinction by aerosols can happen.

Author reply: We have revised it as follows: "… For solar radiation at oxygen A-band and its reference band, they are also attenuated by airmass and aerosol through Rayleigh scattering and aerosol extinction…"

- Line 152: 'between solar and satellite sensors': You mean 'between Sun and satellite sensor'?

Author reply: Yes, we have revised as suggested.

- Line 154: 'layerd' should be 'layered'.

Author reply: We have revised as suggested.

- Line 284: 'and hard to tell directly' should be removed.

Author reply: We have revised as suggested.

- Line 371: 'decrease' should be 'increase' if I understand correctly.

Author reply: Here the "retrieved CTP (with considering cloud penetration)" is smaller than the "baseline CTP (without considering cloud penetration)". Hence, we say "A decrease in retrieved CTP will …" in this sentence.

---

## Author Comment (AC3) · 31 Jul 2020

I can not find a way to submit the revised manuscripts. Thus I want to submit the revised manuscripts as a supplement file temporarily. I already sent an email to the editor for help. I am sorry for that if it brings in any inconvenience to you.

Please also note the supplement to this comment:
https://amt.copernicus.org/preprints/amt-2019-373/amt-2019-373-AC3-supplement.zip